# ROBUST CURRICULUM LEARNING: FROM CLEAN LABEL DETECTION TO NOISY LABEL SELF-CORRECTION

**Tianyi Zhou***, **Shengjie Wang***, **Jeff A. Bilmes**
University of Washington, Seattle
`{tianyizh,wangsj,bilmes}@uw.edu`

## ABSTRACT

Neural network training can easily overfit noisy labels resulting in poor generalization performance. Existing methods address this problem by (1) filtering out the noisy data and only using the clean data for training or (2) relabeling the noisy data by the model during training or by another model trained only on a clean dataset. However, the former does not leverage the features' information of wrongly-labeled data, while the latter may produce wrong pseudo-labels for some data and introduce extra noises. In this paper, we propose a smooth transition and interplay between these two strategies as a curriculum that selects training samples dynamically. In particular, we start with learning from clean data and then gradually move to learn noisy-labeled data with pseudo labels produced by a time-ensemble of the model and data augmentations. Instead of using the instantaneous loss computed at the current step, our data selection is based on the dynamics of both the loss and output consistency for each sample across historical steps and different data augmentations, resulting in more precise detection of both clean labels and correct pseudo labels. On multiple benchmarks of noisy labels, we show that our curriculum learning strategy can significantly improve the test accuracy without any auxiliary model or extra clean data.

## 1 INTRODUCTION

The expressive power and high capacity of deep neural networks (DNNs) result in accurate modeling and promising generalization if provided with sufficient data and clean(correct) labels. However, recent studies show that the training process is fragile and can easily overfit on noisy labels (Zhang et al., 2017), which commonly appear in real-world data since precise annotation is not always available or affordable. Hence, it is important to study the training dynamics affected by imperfect labels and develop robust learning strategies that ideally eliminate the negative impact of noisy labels while fully exploiting the information from all the available data.

Numerous approaches have been developed to address this challenge from various perspectives, e.g., loss correction (Xiao et al., 2015; Vahdat, 2017; Lee et al., 2018; Veit et al., 2017; Li et al., 2017b), robust loss functions (Ghosh et al., 2017; Zhang & Sabuncu, 2018; Wang et al., 2019; Ma et al., 2020) with provable noise tolerance, sample re-weighting (Patrini et al., 2017), curriculum learning (Kumar et al., 2010; Jiang et al., 2018; Guo et al., 2018), model co-teaching (Han et al., 2018), etc. A principal methodology behind a variety of methods is to detect clean labels while discard/downweigh the data with wrong labels, so the model mainly learns from correct labels. A broadly applied criterion is to select the samples with small losses and treat them as clean data. It is inspired by empirical observations that DNNs learn simple patterns first before overfitting on the noisy labels (Zhang et al., 2017; Arpit et al., 2017). Several curriculum learning methods utilize this criterion (Kumar et al., 2010; Jiang et al., 2014), and in each step, select/upweigh samples with small losses. Robust loss functions also suppress the large losses associated with the possibly wrong labels. More recent approaches use mixture models (Arazo et al., 2019) to estimate the distribution of losses for clean and noisy data.

However, the instantaneous loss (i.e., the loss evaluated at the current step) of an individual sample is an unstable signal that can rapidly fluctuate due to DNN training's randomness. The error generated by such an unstable metric accumulates when the selected samples are used to train the model producing the losses. Co-teaching methods alleviate this problem by training two DNNs and using the loss computed on one model to guild the other. Also, as the model changes during training, each sample's loss needs to be re-evaluated even when it is not selected, which requires extra inference

cost. MentorNet (Jiang et al., 2018) and Data Parameters (Saxena et al., 2019) train an extra model to produce the sample weights or selection results without computing the loss. Furthermore, it may not be efficient to repeatedly train the model only on clean data that consistently have small losses, since the model have already learned, well memorized or overfitted to them.

A primary drawback of training only on clean labels detected is that discarding the whole data pairs $(x, y)$ with wrong labels $y$ removes potentially useful information about the data distribution $p(x)$ (Arazo et al., 2019). Hence, there has been growing interest in leveraging noisy data. Loss correction methods aim to correct the predicted class probabilities based on an estimated mislabeling probability between classes. Some other methods seek to relabel them by using the model itself (e.g., bootstrapping loss (Reed et al., 2014)) or another model/mechanism (e.g., directed graphical models, conditional random fields, or CNNs) trained on an additional set of clean data, which, however, is not always available. Self-training and unsupervised learning techniques (Rasmus et al., 2015; Berthelot et al., 2019) have also been employed to generate pseudo labels to replace noisy labels (Arazo et al., 2019). The pseudo labels are optimized together with the model or generated by the model with data augmentations to encourage the output consistency on the same sample's augmentations. Unfortunately, the pseudo labels' quality may vary across different samples and significantly degenerate when the noise ratio is high, or the model fails to produce stable and correct predictions. In such a case, the relabeling error on some samples can be accumulated during training.

In this paper, we address the aforementioned problems of noise-label learning by developing a curriculum learning strategy called *Robust Curriculum Learning* (**RoCL**) that smoothly transitions between two phases: (1) detection and supervised training on clean data; and (2) relabeling and self-supervision on noisy data. Specifically, we train the model for multiple episodes, each starting from phase(1) and gradually moving to phase(2). Unlike existing approaches, we only select samples with accurate given/pseudo labels that are most informative to the current model training. Our data selection criterion takes both the dynamics of per-sample loss and output consistency (across multiple data augmentations) into account. Using an exponential moving average of the loss and consistency over training history, it overcomes the instability of instantaneous losses and does not incur any additional inference cost. In addition, by adjusting a temperature parameter, the criterion can interpolate between the two phases and keep the training focusing on the data that the model mostly needs to improve on, e.g., clean data with unsatisfying output consistency or wrongly-labeled data with accurate pseudo labels. Thus, we can fully exploit both clean and noisy data more efficiently with less risk of introducing extra noise or error accumulation. We further show that our data selection can be derived from a novel optimization formulation for robust curriculum learning. We evaluate our method on multiple noisy learning benchmarks and show that our method outperforms a diverse set of recent noisy-label learning approaches.

## 1.1 RELATED WORK

Early curriculum learning (CL) (Khan et al., 2011; Basu & Christensen, 2013; Spitkovsky et al., 2009; Zhou et al., 2021) seeks an optimized sequence of training samples (i.e., a curriculum, which can be designed by human experts) to improve model performance. Self-paced learning (SPL)(Kumar et al., 2010; Tang et al., 2012a; Supancic III & Ramanan, 2013; Tang et al., 2012b) selects easy samples with smaller losses. It starts with selecting a few samples of small loss and gradually increases the selection size to cover all the training data. Self-paced curriculum learning (Jiang et al., 2015) combines the human expert in CL and loss-adaptation in SPL. SPL with diversity (SPLD) (Jiang et al., 2014) applies a negative group sparse regularization to SPL to promote the diversity of selected samples. Minimax curriculum learning Zhou & Bilmes (2018) promotes the diversity of samples during early learning to encourage exploration and focus on hard samples in later stages.

In the context of robust learning with noisy labels, label correction methods aim to identify the wrong labels and possibly correct them to get more consistent labels for training. Previous work often apply an extra noise model (directed graphical model (Xiao et al., 2015), conditional random fields (Vahdat, 2017), neural network (Lee et al., 2018; Veit et al., 2017), knowledge graph (Li et al., 2017b)) to correct the noisy labels, which often require extra clean data and as well as training/inference of the noise model. Another line of research focuses on loss correction, which modifies the loss or prediction probabilities during training to correct the misinformation from the noisy labels. Patrini et al. (2017) uses two noise transition (backward and forward) matrices to correct the prediction probabilities. Label Smoothing Regularization (Szegedy et al., 2016; Pereyra et al., 2017) alleviates the overfitting to noisy labels by using soft labels instead of one-hot labels. Reed et al. (2014) augments the loss with a notion

of perceptual consistency. Jiang et al. (2018) trains a mentor network to reweigh samples duri'ng the training of a student network. Guo et al. (2018) designs a curriculum by ranking the complexity of data using its distribution density in a feature space. Ren et al. (2018) proposes a meta-learning algorithm that learns to assign weights to samples based on their gradients in training compared to those of validation data, which requires extra clean data. Co-teaching (Han et al., 2018) feeds in the network with the most confident samples of another network to reduce confirmation bias. Amid et al. (2019) generalizes the logistic loss and the exponents in the softmax by applying a temperature to each of them and makes the training more robust to noise. Hu et al. (2019) trains a network on noisy labels in the weakly supervised setting and uses it as a regularization term to improve the training on clean data.

Some approaches focus on designing loss functions that have robust behaviors and provable tolerance to label noise. Ghosh et al. (2017) theoretically proves that the Mean Absolute Error(MAE) is a robust loss. The Generalized Cross Entropy (Zhang & Sabuncu, 2018) uses a negative Box-Cox transformation to obtain a loss function that generalizes MAE and Cross Entropy loss. Wang et al. (2019) proposes a Symmetric Cross Entropy that combines Cross Entropy loss and Reverse Cross Entropy loss. Ma et al. (2020) proposes a loss normalization method and shows that any loss can be made robust to noisy labels.

RoCL shares similar ideas with some CL methods in that RoCL starts with learning easy and clean samples and gradually moves to hard and noisy ones. RoCL is more related to the loss correction approach in noisy-label learning literature as RoCL generates a curriculum dynamically assigning weight (probability) to each sample. RoCL differs from existing methods in: (1) it only selects a subset of informative and reliable labels for training in each epoch; (2) it is a smooth transition not only from clean data to noisy data but also from supervised learning to self-supervision; (3) it runs multiple episodes of the curriculum to avoid getting in a local minimum dominated by a small set of clean/noisy data or a specific type of loss; (4) it does not assume the availability of an extra set of clean data; (5) it does not require extra computation or any modification to the model.

## 2 DYNAMIC PATTERNS OF CLEAN/NOISY LABELS IN TRAINING

### 2.1 LOSS DYNAMICS AND CLEAN LABEL DETECTION

A key challenge for most noise-label learning methods is to design a reliable criterion to select/reweigh clean data and distinguish them from the noisy data, so all the clean data can be fully exploited while most noisy labels are filtered out of the training process. Loss computed at an instantaneous step have been widely used for this purpose according to the observation that the loss on clean data is usually smaller than noisy data. One important reason is that the clean labels are mutually consistent with each other in producing gradient updates, and therefore, the model can fit them better and faster. On the other hand, the noisy labels may contain mutually inconsistent information, creating a form of long-lasting "tug of war" amongst themselves. For example, it can be hard for the model to find consistent visual patterns from images with noisy labels to make the desired predictions. However, instantaneous loss suffers from high variance across training epochs (as shown in the first plot of Figure 1) and is inaccurate for clean data detection under high noise ratio (i.e., the proportion of wrong labels is high) and the randomness of DNN training,

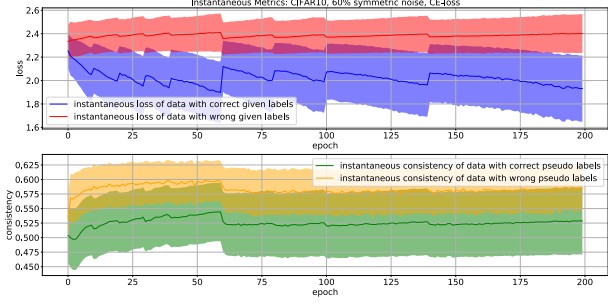

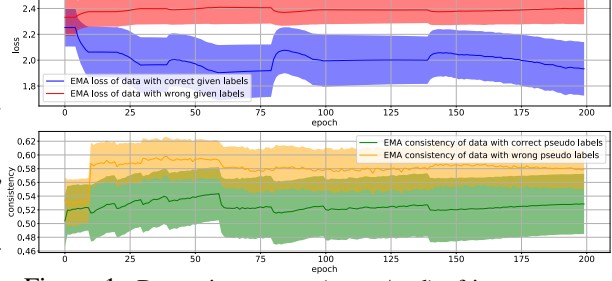

Figure 1: Dynamic patterns (mean±std) of instantaneous metrics (top) and exponential moving average (EMA) metrics (bottom) when applying Alg. 2 that alternates between supervised learning on given labels and self-supervision on pseudo labels. *Larger gap between curves in each plot is better.* Symmetric noise is defined in the beginning of Section 4. We use cross entropy for supervised loss $\ell(\cdot, \cdot)$ in Eq. (1) and 0-1 loss for $\ell(\cdot, \cdot)$ in consistency loss Eq. (2) with $m = 7$.

e.g., random initialization, random data augmentation, etc. Moreover, it needs to evaluate the instantaneous loss for all samples in each step, resulting in extra inference cost on unselected samples.

The dynamic patterns of losses (Zhou et al., 2020b) over the course of training give us a new insight for better clean data detection even when the noise ratio (proportion of wrong labels) is high. In particular, we hypothesize that a sample's label is more likely to be correct if its losses persistently retain low values over training steps. Given a sample $(x_i, y_i)$ with $x_i$ being the features and $y_i$ being the label, we describe its loss dynamics using a simple exponential moving average (EMA) of the instantaneous loss $\ell(f(x_i; \theta_t), y_i)$ (where $f(x_i; \theta_t)$ denotes the model output and $\theta_t$ is the model parameters at step $t$ [*] along the training history, which is defined and computed recursively as

$$l_{t+1}(i) = \begin{cases} \gamma \times \ell(f(x_i; \theta_t), y_i) + (1 - \gamma) \times l_t(i) & \text{if } i \in S_t \\ l_t(i) & \text{else ,} \end{cases} \qquad (1)$$

Where $\gamma \in [0, 1]$ is a discounting factor, $V$ is the set of all $n$ training samples, and $S_t \subseteq V$ is the set of samples selected (by a certain curriculum) for training at epoch $t$. We only update the EMA loss for selected samples using the byproduct $l_t(i)$ of training without requiring extra inference.

In the first and the third plots of Figure 1, we show how the losses and EMA losses associated with clean/noisy data change throughout the training process. Specifically, we train a ResNet34 (He et al., 2016) model on CIFAR10 with $60\%$ of the original labels randomly changed to a wrong class. To avoid quick overfitting to the noise, we train the model for multiple episodes (each composed of several epochs over all data with a cosine annealing learning rate) and alternate between the supervised learning episode that minimizes the cross-entropy loss against the given noisy labels and the self-supervision episode that minimizes the consistency loss Eq.(2) against the pseudo labels. Comparing the shaded areas (std) of instantaneous loss and EMA loss and the gap between curves in the two plots, we see that EMA leads to smaller variance within each group and larger gap between the clean and noisy groups, demonstrating the effectiveness of EMA loss for clean data detection.

## 2.2 Consistency Dynamics and Pseudo Label Selection

Simply removing noisy data $(x, y)$ with wrong label $y$ discards important information about the data distribution $p(x)$ (Arazo et al., 2019). Including all the noisy data for self-supervision, bootstrapping or relabeling can also harm the training since the pseudo labels' quality is not equal across samples and much depends on the model generating them, where the model's predictions may contain errors that can accumulate if adopted for training. Hence, a careful selection of noisy data is necessary. However, without access to a purely clean dataset or a reliable pre-trained model, it is nontrivial to evaluate pseudo labels' correctness. By analyzing the training dynamics of model outputs in the above experiment, we discover that the model output for a sample tends to be an accurate pseudo label if the output remains consistent over training steps and across different augmentations of the sample.

We first define the instantaneous consistency loss of a sample $x_i$ at step $t$ as the discrepancy of the model output $f(x_i; \theta)$ between step $t$ and $t-1$ on $x_i$ and its $m$ data augmentations $\{x_i^{(j)}\}_{j=1}^m$, where the discrepancy can be measured by any loss function $\ell(\cdot, \cdot)$. However, the discrepancy can be small if the models of epoch $t$ and $t-1$ are too similar and make the same errors. Therefore, we use an exponential moving average of the model parameters (according to mean teacher (Tarvainen & Valpola, 2017)) and compute the prediction at step $t-1$ by averaging over multiple data augmentations (according to MixMatch (Berthelot et al., 2019)): $\overline{f}_t(x_i) \triangleq 1/m \sum_{j=1}^m f(x_i^{(j)}; \overline{\theta}_t)$, $\overline{\theta}_t \triangleq \gamma \theta_{t-1} + (1 - \gamma)\overline{\theta}_{t-1}$. Computing pseudo labels on augmented data and a time averaging ensemble of models is commonly-adopted for semi-supervised learning (Sajjadi et al., 2016; Laine & Aila, 2016; Zhou et al., 2020a). The instantaneous consistency loss[†] $\zeta_t(i)$ of sample $x_i$ at step $t$ is then defined as

$$\zeta_t(i) \triangleq \frac{1}{m} \sum_{j=1}^m \ell(f(x_i; \theta_t), \overline{f}_t(x_i^{(j)})), \qquad (2)$$

Which can also be minimized as a consistency loss for $x_i$ in self-supervised learning when no label is given. Similar to the EMA loss in Eq. (1), we define the EMA consistency loss over training history

$$c_{t+1}(i) = \begin{cases} \gamma \times \zeta_t(i) + (1 - \gamma) \times c_t(i) & \text{if } i \in S_t \\ c_t(i) & \text{else.} \end{cases} \qquad (3)$$

Note we use the same $\gamma$ value as in Eq. (1). The EMA consistency loss $c_t(i)$ measures both the time-consistency (Zhou et al., 2020a) over multiple training steps and the spatial consistency over different augmentations of sample $x_i$. If the output prediction is wrong and contradicts other samples'

---

[*]Each step does not strictly refer to a mini-batch: when the subset $S_t$ for step $t$ is larger than the mini-batch size (which is common in practice), we run SGD steps over all mini-batches of the subset for one pass.

[†]This name maybe misleading since it actually measures the "inconsistency", but it is commonly called "consistency" in previous works (Zhu et al., 2017; Berthelot et al., 2019), so we decide to use the same name.

labels, it will be inconsistent over time and across augmentations since it can easily change or flip after the next training step. In the last plot of Figure 1, we report the mean and standard deviation (the middle line and the shaded area) of the EMA consistency loss for two groups of data at each epoch, i.e., the ones with correct pseudo labels and the ones with incorrect pseudo labels. Comparing to the instantaneous consistency loss in the second plot, EMA consistency loss is a more reliable criterion for allocating correct pseudo labels. Thus, we can safely learn the noisy data by using their pseudo labels as training targets and avoid introducing harmful noises.

## 3 ROBUST CURRICULUM LEARNING

In this section, we first introduce the selection criterion for both the clean label detection and pseudo label selection. By adjusting two temperature parameters $\tau_1$ and $\tau_2$, it can smoothly interpolate between the two criteria and control their trade-off. We then show that the criterion is derived from a novel optimization formulation for robust curriculum learning. We finally present the RoCL algorithm.

### 3.1 DATA SELECTION CRITERION

To combine of clean label detection and pseudo label selection, we use the two criteria from the previous sections and apply a temperature parameter to control the preference for small/large loss or consistency loss and their trade-off. We sample $x_i$ at training step $t$ with probability:

$$\mathcal{P}_t(i) = \lambda \times p_t(i) + (1 - \lambda) \times q_t(i), \tag{4}$$

where $p_t(i)$ and $q_t(i)$ are defined as softmax probabilities, i.e.,

$$p_t(i) \triangleq \frac{\exp[\tau_1 l_t(i)]}{\sum_{j=1}^n \exp[\tau_1 l_t(j)]}, \quad q_t(i) \triangleq \frac{\exp[\tau_2 c_t(i)]}{\sum_{j=1}^n \exp[\tau_2 c_t(j)]}, \tag{5}$$

and $\lambda \in [0, 1]$ controls trade-off between them. Here, $p_t(i)$ is a softmax probability computed from EMA losses for clean data detection: samples with smaller (larger) EMA losses and thus clean (noisy) labels tend to have high probability $p_t(i)$ when $\tau_1$ is negative (positive). Similarly, $q_t(i)$ is computed from EMA consistency loss for pseudo label selection: samples with smaller (larger) EMA consistency loss and thus correct (wrong) pseudo labels tend to have high probability $q_t(i)$ when $\tau_2$ is negative (positive). When $\tau_1, \tau_2 = 0$, the probabilities are uniform, and when $\tau_1, \tau_2 \to +\infty/ -\infty$, the probabilities approximate the max (min) operator.

Either $p_t(i)$ or $q_t(i)$ can be independently employed to select or reweigh samples for noise-label learning. However, selecting samples with high probabilities $p_t(i)$ when $\tau_1 \ll 0$ tends to make the training focus on clean data that the model has already learned, which carries limited new information and prevents exploration. Similarly, selecting samples with high probabilities $q_t(i)$ when $\tau_2 \ll 0$ is not informative since the model outputs are consistently correct for those data, and little progress can be made. We can encourage exploration by manipulating the temperature parameters. By setting $\tau_1$ and $\tau_2$ close to zero, we move towards uniform exploration of all data. A more effective strategy is to couple the values of $\tau_1$ and $\tau_2$. For example, a negative $\tau_1$ and a positive $\tau_2$ strengthen the preference for clean data that have not been fully exploited and learned by the model. Alternatively, a positive $\tau_1$ with a negative $\tau_2$ emphasizes the noisy data with correct pseudo labels, so relabeling them provides new information in addition to the clean data.

We apply the data selection criterion of Eq. (4) in each step and gradually change its parameters $(\tau_1, \tau_2, \lambda)$ over the course of training according to the properties discussed above. We start from a negative $\tau_1$ associated with a positive $\tau_2$ and a large $\lambda$ ($p_t(i)$ dominates), then gradually increase $\tau_1$ while decrease $\tau_2$ and $\lambda$, and end with a positive $\tau_1$, a negative $\tau_2$ and a small $\lambda$. In this way, we get a curriculum with a smooth transition between supervised learning of clean data using correct given labels and self-supervision of noisy data using reliable pseudo labels (i.e., minimizing Eq. (2). Moreover, the coupling strategy on $\tau_1$ and $\tau_2$ encourages selecting informative samples that the model mostly needs to improve on, i.e., clean data with inconsistent model outputs or noisy data with correct pseudo labels. In our experiments, we can further reduce the hyperparameters: (1) given the sequence for $\tau_1$ in the curriculum as $\tau_{1:T}$, we can reverse it as the sequence for $\tau_2$, i.e., $\tau_{T:1}$; (2) we can make $\lambda$ monotone increase with $\tau_1$, e.g., setting the initial value $\lambda_1 = a\tau_1 + b$ and ending value $\lambda_T = a\tau_T + b$, solving this linear system for $a$ and $b$, which generate $\lambda_{1:T} = a\tau_{1:T} + b$.

### 3.2 ROBUST CURRICULUM LEARNING AS AN OPTIMIZATION

The data selection criterion is derived from an optimization formulation of our robust curriculum learning (RoCL), in which we aim to minimize a combination of supervised loss and

consistency(self-supervised) loss in the following form.

$$\min_\theta F(\theta) \triangleq \frac{\lambda}{\tau_1} \log \left( \frac{1}{n} \sum_{i=1}^n \exp[\tau_1 \ell(f(x_i; \theta), y_i)] \right) + \frac{1-\lambda}{\tau_2} \log \left( \frac{1}{n} \sum_{i=1}^n \exp[\tau_2 \zeta(i)] \right), \quad (6)$$

where the consistency loss $\zeta(i)$ is defined in Eq. (2) (with $t$ removed). For the first term of Eq. (6), unlike the conventional empirical risk minimization (ERM) with arithmetic average loss, i.e., $1/n \sum_{i=1}^n \ell(f(x_i; \theta), y_i)$, we use LogSumExp loss with an additional temperature parameter (i.e., $\frac{1}{\tau_1} \log \left( \frac{1}{n} \sum_{i=1}^n \exp[\tau_1 \ell(f(x_i; \theta), y_i)] \right)$), so it can approximate both the min-loss (when $\tau_1 \to -\infty$), max-loss (when $\tau_1 \to +\infty$), and any interpolation between them, where the min-loss focuses on the easiest samples with the smallest losses and the max-loss focuses on the hardest ones. Note it reduces to the arithmetic average loss when $\tau \to 0$. It is called "tilted loss" in a recent work (Li et al., 2020), which shows several intriguing properties in different learning settings. The second term of Eq. (6) focuses on the consistency loss $\zeta(i)$. We use $\lambda$ to control the trade-off between the supervised loss and the consistency loss. By simple algebra, the gradient of $F(\theta)$ at step $t$ is

$$\nabla_\theta F(\theta_t) = \lambda \sum_{i=1}^n p_t'(i) \nabla_\theta \ell(f(x_i; \theta_t), y_i) + (1-\lambda) \sum_{i=1}^n q_t'(i) \nabla_\theta \zeta_t(i)$$

$$= \sum_{i=1}^n \mathcal{P}_t'(i) \left[ \frac{\lambda p_t'(i)}{\mathcal{P}_t'(i)} \nabla_\theta \ell(f(x_i; \theta_t), y_i) + \frac{(1-\lambda) q_t'(i)}{\mathcal{P}_t'(i)} \nabla_\theta \zeta_t(i) \right] = \mathbb{E}_{i \sim \mathcal{P}_t'(i)} G_t(i), \quad (7)$$

where

$$G_t(i) \triangleq \frac{\lambda p_t'(i)}{\mathcal{P}_t'(i)} \nabla_\theta \ell(f(x_i; \theta_t), y_i) + \frac{(1-\lambda) q_t'(i)}{\mathcal{P}_t'(i)} \nabla_\theta \zeta_t(i). \quad (8)$$

Here, $p_t'(i)$ and $q_t'(i)$ are similar to $p_t(i)$ and $q_t(i)$ defined in Eq. (5) except that the EMA loss and EMA consistency loss are replaced by their instantaneous counterparts respectively, i.e., $p_t'(i) \triangleq \frac{\exp[\tau_1 \ell(f(x_i; \theta_t), y_i)]}{\sum_{j=1}^n \exp[\tau_1 \ell(f(x_j; \theta_t), y_j)]}$, $q_t'(i) \triangleq \frac{\exp[\tau_2 \zeta_t(i)]}{\sum_{j=1}^n \exp[\tau_2 \zeta_t(j)]}$. Similarly, we also denote $\mathcal{P}_t'(i) = \lambda \times p_t'(i) + (1-\lambda) \times q_t'(i)$. Note in Section 2, we already discussed that the EMA metrics are better alternatives to the instantaneous metrics when used for data selection in noisy-label learning. In our experiments, we use the EMA metrics $\{p_t(i), q_t(i), \mathcal{P}_t(i)\}$ instead of the instantaneous ones $\{p_t'(i), q_t'(i), \mathcal{P}_t'(i)\}$. For every training step, an unbiased estimator of the gradient in Eq. (7) can be achieved by drawing a subset of samples $S_t$ according to $\mathcal{P}_t(i)$ and averaging their gradients $G_t(i)$ in Eq. (8).

## 3.3 ROBUST CURRICULUM LEARNING ALGORITHM

We describe our curriculum learning algorithm RoCL in Alg. 1 based on the data selection criterion in Section 3.1 and the optimization formulation in Section 3.2. We denote the update of $\theta_t$ produced by the adopted optimizer as $h(\nabla_\theta F(\theta_t), \eta)$, where $\eta$ contains all hyperparameters of the optimizer at step $t$, e.g., the learning rate. We denote $\ell_t(i)$ as a shorthand notation for $\ell(f(x_i; \theta_{t-1}), y_i)$. We apply a warm starting episode of a few epochs (e.g., 5-10) over all the data and given labels with label smoothing to obtain stable EMA metrics. After that, we apply multiple episodes of curriculum learning, each including a sequence of steps following the curriculum at the end of Section 3.1 for data selection per step (Line 6-14). We repeat the transition between clean data learning to noisy data learning for $K$ episodes to avoid getting trapped in a local

---

**Algorithm 1** Robust Curriculum Learning (RoCL)

1: **input:** $\{(x_i, y_i)\}_{i=1}^n$, $h(\cdot; \eta)$, $\ell(\cdot, \cdot)$, $f(\cdot; \theta)$, $T_{0:K}$; $\tau_1 < 0, \tau_T > 0$; $\lambda_1, \lambda_T \in [0, 1]$; $\gamma, \gamma_b \in [0, 1]$
2: **initialize:** $\theta_0, b_0 \in (0, n)$, $l_0(i) = c_0(i) = 0 \; \forall i \in [n]$
3: **for** $k \in \{0, \cdots, K\}$ **do**
4:     Schedule $\tau_{1:T_k}$ and $\lambda_{1:T_k}$ by Eq. (9)-(10);
5:     **for** $t' \in \{1, \cdots, T_k\}$ **do**
6:       $t \leftarrow t' + T_{k-1}$;
7:       **if** $k = 0$ **then**
8:         $S_t \leftarrow [n]$;
9:         $\theta_t \leftarrow \theta_{t-1} + h\left(\nabla_\theta \frac{1}{n} \sum_{i=0}^n \ell_t(i); \eta\right)$;
10:       **else**
11:         Draw a subset $S_t \subseteq [n]$ of $b_k$ samples according to probability $\mathcal{P}_t$ in Eq. (4) with $\tau_1 = \tau_{t'}, \tau_2 = \tau_{T_k - t'}, \lambda = \lambda_{t'}$;
12:         $\theta_t \leftarrow \theta_{t-1} + h\left(\frac{1}{b_k} \sum_{i \in S_t} G_t(i); \eta\right)$ (Eq. (8));
13:       **end if**
14:       Update $l_{t+1}(i)$ and $c_{t+1}(i)$ by Eq. (1) and Eq. (3);
15:     **end for**
16:     $b_{k+1} \leftarrow (1 + \gamma_b) \times b_k$;
17: **end for**

---

minimum dominated by a small set of clean/noisy data or a specific type of loss. It also reinforces

the memorization of clean labels and correct pseudo labels learned in previous episodes. Moreover, under the coupling strategy of $\tau_1$ and $\tau_2$, each episode is encouraged to explore the clean/noisy data that the previous episode fails to learn. Considering the undertrained model (producing inaccurate pseudo labels) and the relatively high variance of the EMA metrics at the earlier episodes, we start from a small budget for the selected subset size and gradually increase in later episodes.

To generate the whole schedule of $\tau_{1:T}$ in each episode, we can apply any monotone interpolation between $\tau_1$ and $\tau_T$ whose values are predefined. Let $g : \mathcal{R} \mapsto [-\sigma, \sigma]$ be an invertible monotone continuous function. We define the interpolation between $\tau_1$ and $\tau_T$ as follows, $\forall t \in [T]$,

$$\tau_t = \frac{\tau_T - \tau_m}{\sigma} \times \left[ g(\sigma_t) - \frac{g(-\sigma) + g(\sigma)}{2} \right] + \tau_m, \ \sigma_t = g^{-1}(-\sigma) + \frac{2t}{T} g^{-1}(\sigma), \ \tau_m = \frac{\tau_1 + \tau_T}{2}. \quad (9)$$

Note the $g(\sigma_t)$ produces interpolation values between $[-\sigma, \sigma]$ that correspond to $T$ evenly spaced input $\sigma_t \in [g^{-1}(-\sigma), g^{-1}(\sigma)]$. In our curriculum for each episode, we need to keep a high quality of the selected clean (pseudo) labels in earlier (later) stages and make the exploration stages in between shorter since their selected labels contain more noise. Therefore, we choose "s"-shaped functions such as $\tanh$ or the logistic function for the interpolation. In this paper, we use $g(\cdot) = \tanh(\cdot)$ and pick $\sigma = 0.95$. We illustrate Eq. (9) and visualize our choice of $g(\cdot)$ and the resulting $\tau_t$ in Figure 6 (Appendix). The corresponding schedule for $\lambda$ can then be defined as an affine transformation of $\tau_{1:T}$:

$$\forall t \in [T], \lambda_t = a_\lambda(\tau_t - \tau_1) + \lambda_1, a_\lambda = \frac{\lambda_T - \lambda_1}{\tau_T - \tau_1}. \quad (10)$$

## 4 EXPERIMENTS

We evaluate RoCL with other approaches for noisy-label learning on three widely used benchmarks, i.e., CIFAR10/100 with two types of synthetic noises (i.e., symmetric and asymmetric), and mini-WebVision (Li et al., 2017a) (the first 50 classes) containing unknown noises from web labels. Symmetric noise flips each label randomly to an incorrect class with probability $\rho$ (i.e., noise rate), and our experiments cover $\rho = \{0.4, 0.6, 0.8\}$. Asymmetric noise flips the labels within a specific set of classes. For CIFAR10, flipping TRUCK→AUTOMOBILE, BIRD→AIRPLANE, DEER→HORSE, CAT→DOG. In CIFAR100, the 100 classes are grouped into 20 super-classes with each has 5 sub-classes, we then flip each class within the same super-class to the next in a circular fashion with probability $\rho$. Our experiments cover $\rho = \{0.2, 0.3, 0.4\}$.

Table 1: Accuracy (%) evaluated on WebVision and ILSVRC2012 validation sets for DNNs trained by noisy-label learning methods on mini-WebVision training set (first 50 classes), which contains **real-world web-label noises**.

| Val. set | WebVision | | ILSVRC2012 | |
|---|---|---|---|---|
| Accuracy | Top-1 | Top-5 | Top-1 | Top-5 |
| F-correct [+*] | 61.12 | 82.68 | 57.36 | 82.36 |
| Decoupling [**] | 62.54 | 84.74 | 58.26 | 82.26 |
| Co-teaching [*] | 63.58 | 85.20 | 61.48 | 84.70 |
| MentorNet [**] | 63.00 | 81.40 | 57.80 | 79.92 |
| MentorMix [*‡*] | 76.00 | 90.20 | 72.90 | 91.10 |
| D2L [*] | 62.68 | 84.00 | 57.80 | 81.36 |
| INCV [*] | 65.24 | 85.34 | 61.60 | 84.98 |
| RoCL (ours) [‡*†ℓ] | **80.04** | **92.68** | **75.81** | **92.28** |

**Practical Modifications** In the experiments, we follow previous work and apply the techniques below. We will present an ablation study of their effectiveness in Table 5.

- We apply the **class-balance regularization** used in (Tanaka et al., 2018), which prevents the model from predicting the same class for all the samples within a mini-batch $B$. We add $(1/B) \sum_{i \in B} \ell(f(x_i; \theta), 1/C \cdot \mathbf{1})$ (where $C$ is the number of classes) to the objective for a mini-batch $B$ with regularization weight of 1.

- We apply **label smoothing** whose effectiveness in noisy-label learning has been studied in (Lukasik et al., 2020). We modify each one-hot label $y_i$ to be $\bar{y}_i \leftarrow (1 - \alpha)y_i + \alpha/C$ (e.g., we use $\alpha = 0.5$).

- We apply **Mix-Up** (Zhang et al., 2018) to all the selected data. However, Mix-Up of two (soft) pseudo labels can significantly increase the entropy of the mixed label if both pseudo labels are under-confident. Hence, we apply a curriculum to the beta distribution's parameter $\alpha$ of Mix-Up and gradually reduce it (e.g., from $8.0$ to $0.2$ in our experiments) within each episode.

**Hyperparameter Setting** We apply RoCL to train ResNet34 on CIFAR10/100 and ResNet50 on WebVision, which are the most widely used models in other baseline papers. We apply SGD with momentum of 0.9, weight decay of $10^{-4}$ and cosine annealing learning rate in each training episode. The initial learning rate is set to 0.1 for CIFAR10/100 and 1.0 for WebVision. In all RoCL experiments, we apply $T_0 = 10$ warm starting epochs followed by $K = 10$ episodes of curriculum learning,

whose lengths start from $T_1 = 10$ and increase by 10 for every episode afterwards. We initialize the subset size $b_0 = 0.2n$ and set $\gamma = \gamma_b = 0.1$, which are common choices for discounting/augmenting factors. We use Cubuk et al. (2020) for data augmentations. We did not heavily tune $\lambda_1, \lambda_T$ and $\tau_1, \tau_T$ and followed a principle that the resulting curriculum should have a transition from supervised learning on clean data to self-supervised learning on noisy data with correct pseudo labels.

- For $\lambda$, we start from $\lambda_1$ close to 1 and end with $\lambda_T$ close to 0 because our curriculum is a transition from supervised learning ($\lambda = 1$) to self-supervised learning ($\lambda = 0$).

- As explained in Section 3.1, our curriculum requires $\tau_1$ for $p_t(i)$ changing from negative to positive values and an inverse sequence for $\tau_2$ in $q_t(i)$ to gain the above transition and encourage learning on more informative samples. Hence, we set the starting value $\tau_1$ to be negative and $\tau_T$ to be positive for the sequence $\tau_{1:T}$.

- We set their exact values based on observations in Figure 1: clean data detection is easier but the detection of correct pseudo-labels is harder. So we can be more confident on the former than the latter and set the starting $\tau_1$ larger than $\tau_T$ in magnitude. For the same reason, we set the starting value $\lambda_1$ to be closer to 1 than the ending value $\lambda_T$ to 0. In experiments, we tried $\tau_1 = \{-4, -3\}$ and $\tau_T = \{1, 2\}$ and finally chose $\tau_1 = -4$ and $\tau_T = 1$ since this choice performs consistently well on all experiments, though it might not be the best choice for all; we set $\lambda_1 = 0.9$ and did not try other values; we tried $\lambda_T = \{0.1, 0.2, 0.3\}$ and on some experiments the first two choices lead to slightly worse performance, so we chose $\lambda_T = 0.3$.

We compare RoCL to the following baselines: F-correct (Patrini et al., 2017), Decoupling (Malach & Shalev-Shwartz, 2017), Co-teaching (Han et al., 2018), D2L (Ma et al., 2018), INCV (Chen et al., 2019), MD-DYR-SH (Arazo et al., 2019), MentorNet (Jiang et al., 2018), MentorMix (Jiang et al., 2020), O2U-net (Huang et al., 2019), RoG+D2L (Lee et al., 2019), PENCIL (Yi & Wu, 2019), GCE (Zhang & Sabuncu, 2018), SCE (Wang et al., 2019), NFL/NCE variants (Ma et al., 2020), and Bootstrap (Reed et al., 2014). To better compare and categorize different baseline methods, we use the following symbols to denote the techniques used: $+$ for

Table 2: Test accuracy (%) of **RoCL** applied with different loss functions on CIFAR10 corrupted by $\{60\%, 80\%\}$ **symmetric(uniform)** noises (CE-cross entropy).

| Noise Rate | 60% | 80% |
|---|---|---|
| CE | $90.22 \pm 0.24$ | $77.47 \pm 0.67$ |
| GCE | $89.30 \pm 0.68$ | $79.84 \pm 1.12$ |
| SCE | $\mathbf{92.06 \pm 0.23}$ | $74.25 \pm 0.86$ |
| NFL+MAE | $88.73 \pm 0.47$ | $\mathbf{85.76 \pm 0.26}$ |
| NFL+RCE | $87.68 \pm 0.35$ | $80.09 \pm 0.41$ |
| NCE+MAE | $90.37 \pm 0.43$ | $82.16 \pm 0.93$ |
| NCE+RCE | $88.03 \pm 0.39$ | $80.33 \pm 0.80$ |

additional clean training data; $*$ for training additional auxiliary models; $\ddagger$ for using mixup; $\star$ for using data augmentations; $\dagger$ for class-balance regularization; $\wr$ for label-smoothing. We report the results and comparisons to baselines in three tables: real-world noise in Table. 1, symmetric noise in Table. 3 and asymmetric noise in Table. 4. RoCL achieves the best performance in every setting, and for most of the cases, improves upon the existing methods by large margins. The closest rival to RoCL is MentorMix, which utilizes MentorNet and Mix-Up to assign weights to each sample. We note that MentorMix requires training of an extra mentor network to generate the sample weights, while RoCL is more flexible and only makes changes to the training process without modifying the model. Table. 2 reports RoCL's performance when applied with different loss functions on

Table 3: Test accuracy (%) of noisy-label learning methods on CIFAR10/100 corrupted by **symmetric(uniform)** label noises of different levels. All the baselines' results are from the original papers or the following-up works. There are two formats of these reported results: "mean±variance" of 5 trials and single-trial accuracy.

| Dataset | CIFAR10 | | | CIFAR100 | | |
|---|---|---|---|---|---|---|
| Noise Rate | 40% | 60% | 80% | 40% | 60% | 80% |
| MD-DYR-SH $^{**\ddagger\dagger}$ | 92.3 | 86.1 | 74.1 | 70.1 | 59.5 | 39.5 |
| MentorNet $^{**}$ | 91.2 | 74.2 | 60.0 | 68.5 | 61.2 | 35.5 |
| MentorMix $^{*\ddagger*}$ | 94.2 | 91.3 | 81.0 | 71.3 | 64.6 | 41.2 |
| O2U-net $^{*}$ | 90.3 | - | 43.4 | 69.2 | - | 39.4 |
| RoG+D2L $^{**}$ | 87.0 | 78.0 | - | 64.9 | 40.6 | - |
| PENCIL $^{*}$ | - | - | - | $69.12 \pm 0.62$ | $57.79 \pm 3.86$ | fail |
| GCE $^{*}$ | $87.62 \pm 0.26$ | $82.70 \pm 0.23$ | $67.92 \pm 0.60$ | $62.64 \pm 0.33$ | $54.04 \pm 0.56$ | $29.60 \pm 0.51$ |
| SCE $^{*}$ | $85.34 \pm 0.07$ | $80.07 \pm 0.02$ | $53.81 \pm 0.27$ | $53.69 \pm 0.07$ | $41.47 \pm 0.04$ | $15.00 \pm 0.04$ |
| NFL+MAE $^{*}$ | $83.81 \pm 0.06$ | $76.36 \pm 0.31$ | $45.23 \pm 0.52$ | $58.18 \pm 0.08$ | $46.10 \pm 0.50$ | $24.78 \pm 0.82$ |
| NCE+RCE $^{*}$ | $86.02 \pm 0.09$ | $79.78 \pm 0.50$ | $52.71 \pm 1.90$ | $59.48 \pm 0.56$ | $47.12 \pm 0.62$ | $25.80 \pm 1.12$ |
| RoCL (ours) $^{\ddagger*\dagger\wr}$ | $\mathbf{94.55 \pm 0.12}$ | $\mathbf{92.06 \pm 0.23}$ | $\mathbf{85.76 \pm 0.26}$ | $\mathbf{74.64 \pm 0.43}$ | $\mathbf{66.79 \pm 0.58}$ | $\mathbf{53.89 \pm 0.62}$ |

Table 4: Test accuracy (%) of noisy-label learning methods on CIFAR10/100 corrupted by **asymmetric(class-dependent)** noises of 3 levels. All the baselines' results are from the original papers or the following-up works.

| Dataset | CIFAR10 | | | CIFAR100 | | |
|---|---|---|---|---|---|---|
| Noise Rate | 20% | 30% | 40% | 20% | 30% | 40% |
| PENCIL [*] | 92.43 | 91.84 | 91.01 | $74.70 \pm 0.56$ | $72.52 \pm 0.38$ | $63.61 \pm 0.23$ |
| Bootstrap [*] | $86.57 \pm 0.08$ | $84.86 \pm 0.05$ | $79.76 \pm 0.07$ | $63.44 \pm 0.35$ | $63.18 \pm 0.35$ | $62.08 \pm 0.22$ |
| F-correct [+*] | $89.09 \pm 0.47$ | $86.79 \pm 0.36$ | $83.55 \pm 0.58$ | $42.46 \pm 2.16$ | $38.13 \pm 2.97$ | $34.44 \pm 1.93$ |
| GCE [*] | $86.07 \pm 0.31$ | $80.78 \pm 0.21$ | $74.98 \pm 0.32$ | $59.99 \pm 0.83$ | $53.99 \pm 0.29$ | $41.49 \pm 0.79$ |
| SCE [*] | $83.92 \pm 0.07$ | $79.70 \pm 0.27$ | $78.20 \pm 0.03$ | $58.22 \pm 0.47$ | $49.85 \pm 0.91$ | $42.19 \pm 0.19$ |
| NFL+MAE [*] | $86.81 \pm 0.32$ | $83.91 \pm 0.34$ | $77.16 \pm 0.10$ | $63.10 \pm 0.22$ | $56.19 \pm 0.61$ | $43.51 \pm 0.42$ |
| NCE+RCE [*] | $88.56 \pm 0.17$ | $85.58 \pm 0.44$ | $79.59 \pm 0.40$ | $62.68 \pm 0.79$ | $57.82 \pm 0.41$ | $46.79 \pm 0.96$ |
| RoCL (ours) [‡*†ɬ] | $\mathbf{95.38 \pm 0.21}$ | $\mathbf{94.19 \pm 0.28}$ | $\mathbf{92.31 \pm 0.35}$ | $\mathbf{80.03 \pm 0.34}$ | $\mathbf{77.59 \pm 0.45}$ | $\mathbf{73.28 \pm 0.83}$ |

CIFAR10 under high noise rates, i.e., 60% and 80%. We observe significant improvements over their performance without using RoCL in Table. 3. It indicates that RoCL is compatible with any loss function and can further enhance their performance.

To analyze the effect of each component in RoCL, we conduct a thorough ablation study of 10 variants of RoCL, each removing/changing one component of the original RoCL. In Table 5, we report their test accuracies on CIFAR10/100 with noise rates of $\{60\%, 80\%\}$. In Figure 7-10 in Appendix, we report how their test accuracies change during the training to study their learning efficiency and convergence. Among them, "no ClassBalance" removes the class-balance regularization; "no RandAugment" replaces the strong data augmentation RandAugment Cubuk et al. (2020) with random crop and random horizontal flip; "no RandSampling" replaces the weighted sampling in Line 11 of Algorithm 1 by selecting the top-$b_k$ samples with the largest $\mathcal{P}_t(i)$; "no EMA metrics" replaces EMA loss and EMA

Table 5: **Ablation study:** Test accuracy (%) of RoCL variants with one part removed/changed when applied to CIFAR10/100 corrupted by **symmetric(uniform)** label noise.

| Dataset | CIFAR10 | | CIFAR100 | |
|---|---|---|---|---|
| Noise Rate | 60% | 80% | 60% | 80% |
| RoCL: no MixUp | 92.98 | **88.18** | **69.72** | **58.72** |
| RoCL: no LabelSmooth | 91.94 | 85.05 | 62.92 | 42.95 |
| RoCL: no ClassBalance | **93.08** | 74.91 | 62.66 | 43.94 |
| RoCL: no RandAugment | 86.59 | 72.35 | 64.84 | 44.06 |
| RoCL: no RandSampling | 92.31 | 85.99 | 64.09 | 57.00 |
| RoCL: no EMA metrics | 92.84 | 87.79 | 65.99 | 53.10 |
| RoCL: $p_t(i) = 1/n$ | 92.42 | 86.05 | 62.69 | 44.35 |
| RoCL: $q_t(i) = 1/n$ | 92.59 | 86.93 | 64.71 | 50.79 |
| RoCL: $p_t(i) = q_t(i) = 1/n$ | 92.07 | 85.77 | 64.18 | 47.88 |
| RoCL$_{Base}$: no curriculum | 87.83 | 66.93 | 61.84 | 41.92 |
| RoCL: original version | 92.82 | 88.00 | 66.79 | 54.22 |
| MentorMix: +RandAugment | 85.45 | 20.68 | 52.70 | 8.02 |
| MentorMix: +RandAugment-MixUp | 84.31 | 38.21 | 58.31 | 8.18 |
| MentorMix: original version | 91.30 | 81.00 | 64.60 | 41.20 |

consistency loss with their instantaneous counterparts; "$p_t(i) = 1/n$" samples the clean data using uniform probabilities; "$q_t(i) = 1/n$" samples the correct pseudo-labels using uniform probabilities; "$p_t(i) = q_t(i) = 1/n$" uses uniform probabilities for both. Note for the final three variants, we still have the curriculum of $\lambda$. We keep the same hyperparameter settings as the original RoCL. We give brief conclusions here and leave a detailed analysis to Appendix: (1) Except RoCL$_{Base}$ in Algorithm 2, "no RandAugment" and "no ClassBalance", most variants perform similarly as the original RoCL and outperform the previous SoTA achieved by MentorMix. The removed components are more important under higher noise rates. (2) RoCL$_{Base}$ removes our proposed curriculum and preserves all other techniques but shows significant degradation on accuracies, indicating that the curriculum is essential to RoCL's appealing performance. (3) A strong data augmentation is critical to effective self-supervision and accurate EMA consistency loss estimation in RoCL, while a weak one may lead to error accumulation. However, applying RandAugment in MentorMix degrades its original performance. (4) Class-balance regularization is only important under very high noise rates. (5) Removing Mix-Up can improve RoCL's performance since it damages information when mixing soft pseudo labels. (6) Compared to other variants, "no RandSampling" or "no EMA metrics" causes less degeneration on the final accuracies but can slow down the convergence and learning speed in the early stages when exploration is insufficient. (7) Changing $p_t(i), q_t(i)$ or both to uniform probabilities reduces the final accuracies in all cases and significantly slows down the learning process.

## 5 CONCLUSION

We propose a novel curriculum learning method RoCL for robust learning under label noises. RoCL features a smooth transition from learning with clean data to noisy data, and from learning with supervised loss to self-supervised loss. Based on observations of training dynamics, RoCL can select samples with reliable labels/pseudo labels and most informative to training. RoCL does not require availability of extra clean data or training of extra auxiliary models. On multiple benchmarks of noisy label learning, RoCL significantly improves upon existing baselines.

## ACKNOWLEDGMENTS

This research is based upon work supported by the National Science Foundation under Grant No. IIS-1162606, the National Institutes of Health under award R01GM103544, and by a Google, a Microsoft, and an Intel research award. It is also supported by the CONIX Research Center, one of six centers in JUMP, a Semiconductor Research Corporation (SRC) program sponsored by DARPA. Some GPUs used to produce the experimental results are donated by NVIDIA. We would like to thank ICLR area chairs and anonymous reviewers for their efforts in reviewing this paper and their constructive comments! We also thank all the MELODI lab members for their helpful discussions and feedback.

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

## A    APPENDIX

### A.1    RoCL$_{Base}$ (NO CURRICULUM) IN SECTION 2 AND FIGURE 2-10

---

**Algorithm 2** RoCL$_{Base}$ (no curriculum)

---

1: **input:** $\{(x_i, y_i)\}_{i=1}^n, h(\cdot; \eta), \ell(\cdot, \cdot), f(\cdot; \theta), T_{0:K}; \gamma \in [0, 1]$
2: **initialize:** $\theta_0, l_0(i) = c_0(i) = 0 \; \forall i \in [n], T_{-1} = 0$
3: **for** $k \in \{0, \cdots, K\}$ **do**
4:     **for** $t' \in \{1, \cdots, T_k\}$ **do**
5:         $t \leftarrow t' + T_{k-1}$;
6:         $S_t \leftarrow [n]$;
7:         **if** $k\%2 = 0$ **then**
8:             $\theta_t \leftarrow \theta_{t-1} + h\left(\nabla_\theta \frac{1}{n} \sum_{i=0}^n \ell_t(i); \eta\right)$; {supervised learning using given labels}
9:             Update $l_{t+1}(i)$ by Eq. (1); {update EMA loss}
10:         **else**
11:             $\theta_t \leftarrow \theta_{t-1} + h\left(\nabla_\theta \frac{1}{n} \sum_{i=0}^n \zeta_t(i); \eta\right)$; {self-supervised learning using pseudo labels}
12:             Update $c_{t+1}(i)$ by Eq. (3); {update EMA consistency loss}
13:         **end if**
14:     **end for**
15: **end for**

---

Note the EMA metrics in line 9 and line 12 are not used for training in RoCL$_{Base}$. They have been updated and recorded for the purpose of empirical study presented in Section 2.

### A.2    ADDITIONAL EXPERIMENTS

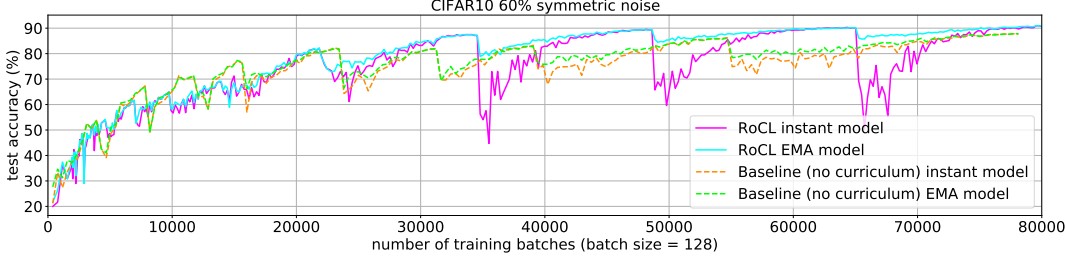

Figure 2: RoCL (Algorithm 1) vs. RoCL$_{Base}$ without any curriculum (Algorithm 2 in Appendix) during the training of ResNet34 on **CIFAR10** containing **60% symmetric noises** on labels.

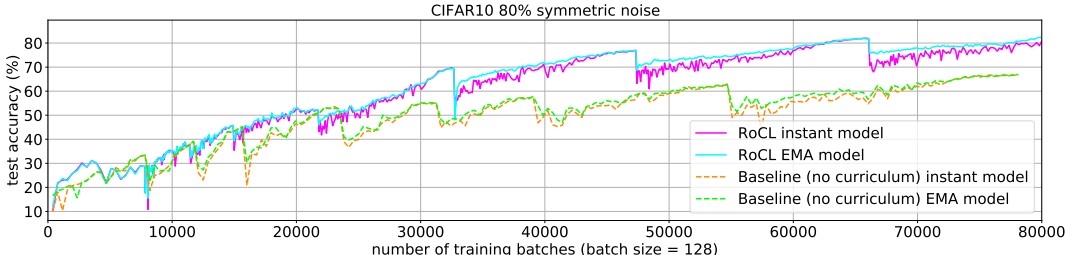

Figure 3: RoCL (Algorithm 1) vs. RoCL$_{Base}$ without any curriculum (Algorithm 2 in Appendix) during the training of ResNet34 on **CIFAR10** containing **80% symmetric noises** on labels.

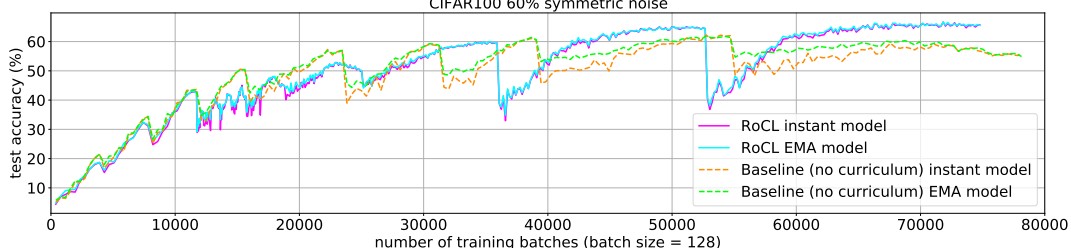

Figure 4: RoCL (Algorithm 1) vs. RoCL$_{Base}$ without any curriculum (Algorithm 2 in Appendix) during the training of ResNet34 on **CIFAR100** containing **60% symmetric noises** on labels.

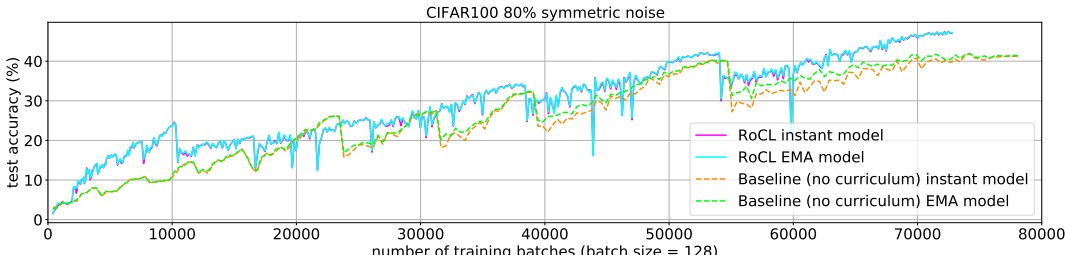

Figure 5: RoCL (Algorithm 1) vs. RoCL$_{Base}$ without any curriculum (Algorithm 2 in Appendix) during the training of ResNet34 on **CIFAR100** containing **80% symmetric noises** on labels.

Table 6: Extended version of Table 3 with two more baselines: NFL+RCE and NCE+MAE.

| Dataset | CIFAR10 | | | CIFAR100 | | |
|---|---|---|---|---|---|---|
| Noise Rate | 40% | 60% | 80% | 40% | 60% | 80% |
| MD-DYR-SH | 92.3 | 86.1 | 74.1 | 70.1 | 59.5 | 39.5 |
| MentorNet | 91.2 | 74.2 | 60.0 | 68.5 | 61.2 | 35.5 |
| MentorMix | 94.2 | 91.3 | 81.0 | 71.3 | 64.6 | 41.2 |
| O2U-net | 90.3 | - | 43.4 | 69.2 | - | 39.4 |
| RoG+D2L | 87.0 | 78.0 | - | 64.9 | 40.6 | - |
| PENCIL | - | - | - | $69.12 \pm 0.62$ | $57.79 \pm 3.86$ | fail |
| GCE | $87.62 \pm 0.26$ | $82.70 \pm 0.23$ | $67.92 \pm 0.60$ | $62.64 \pm 0.33$ | $54.04 \pm 0.56$ | $29.60 \pm 0.51$ |
| SCE | $85.34 \pm 0.07$ | $80.07 \pm 0.02$ | $53.81 \pm 0.27$ | $53.69 \pm 0.07$ | $41.47 \pm 0.04$ | $15.00 \pm 0.04$ |
| NFL+MAE | $83.81 \pm 0.06$ | $76.36 \pm 0.31$ | $45.23 \pm 0.52$ | $58.18 \pm 0.08$ | $46.10 \pm 0.50$ | $24.78 \pm 0.82$ |
| NFL+RCE | $86.05 \pm 0.12$ | $79.78 \pm 0.13$ | $55.06 \pm 1.08$ | $58.20 \pm 0.31$ | $46.30 \pm 0.45$ | $25.16 \pm 0.55$ |
| NCE+MAE | $84.19 \pm 0.43$ | $77.61 \pm 0.05$ | $49.62 \pm 0.72$ | $59.22 \pm 0.36$ | $48.06 \pm 0.34$ | $25.50 \pm 0.76$ |
| NCE+RCE | $86.02 \pm 0.09$ | $79.78 \pm 0.50$ | $52.71 \pm 1.90$ | $59.48 \pm 0.56$ | $47.12 \pm 0.62$ | $25.80 \pm 1.12$ |
| RoCL (ours) [‡⋆†ℓ] | $\mathbf{94.55 \pm 0.12}$ | $\mathbf{92.06 \pm 0.23}$ | $\mathbf{85.76 \pm 0.26}$ | $\mathbf{74.64 \pm 0.43}$ | $\mathbf{66.79 \pm 0.58}$ | $\mathbf{53.89 \pm 0.62}$ |

Table 7: Extended version of Table 4 with two more baselines: NFL+RCE and NCE+MAE.

| Dataset | CIFAR10 | | | CIFAR100 | | |
|---|---|---|---|---|---|---|
| Noise Rate | 20% | 30% | 40% | 20% | 30% | 40% |
| PENCIL | 92.43 | 91.84 | 91.01 | $74.70 \pm 0.56$ | $72.52 \pm 0.38$ | $63.61 \pm 0.23$ |
| Bootstrap | $86.57 \pm 0.08$ | $84.86 \pm 0.05$ | $79.76 \pm 0.07$ | $63.44 \pm 0.35$ | $63.18 \pm 0.35$ | $62.08 \pm 0.22$ |
| F-correct | $89.09 \pm 0.47$ | $86.79 \pm 0.36$ | $83.55 \pm 0.58$ | $42.46 \pm 2.16$ | $38.13 \pm 2.97$ | $34.44 \pm 1.93$ |
| GCE | $86.07 \pm 0.31$ | $80.78 \pm 0.21$ | $74.98 \pm 0.32$ | $59.99 \pm 0.83$ | $53.99 \pm 0.29$ | $41.49 \pm 0.79$ |
| SCE | $83.92 \pm 0.07$ | $79.70 \pm 0.27$ | $78.20 \pm 0.03$ | $58.22 \pm 0.47$ | $49.85 \pm 0.91$ | $42.19 \pm 0.19$ |
| NFL+MAE | $86.81 \pm 0.32$ | $83.91 \pm 0.34$ | $77.16 \pm 0.10$ | $63.10 \pm 0.22$ | $56.19 \pm 0.61$ | $43.51 \pm 0.42$ |
| NFL+RCE | $88.73 \pm 0.29$ | $85.74 \pm 0.22$ | $79.27 \pm 0.43$ | $63.12 \pm 0.41$ | $54.72 \pm 0.38$ | $42.97 \pm 1.03$ |
| NCE+MAE | $86.44 \pm 0.23$ | $83.98 \pm 0.52$ | $78.23 \pm 0.42$ | $62.38 \pm 0.60$ | $58.02 \pm 0.48$ | $47.22 \pm 0.30$ |
| NCE+RCE | $88.56 \pm 0.17$ | $85.58 \pm 0.44$ | $79.59 \pm 0.40$ | $62.68 \pm 0.79$ | $57.82 \pm 0.41$ | $46.79 \pm 0.96$ |
| RoCL (ours) | $\mathbf{95.38 \pm 0.21}$ | $\mathbf{94.19 \pm 0.28}$ | $\mathbf{92.31 \pm 0.35}$ | $\mathbf{80.03 \pm 0.34}$ | $\mathbf{77.59 \pm 0.45}$ | $\mathbf{73.28 \pm 0.83}$ |

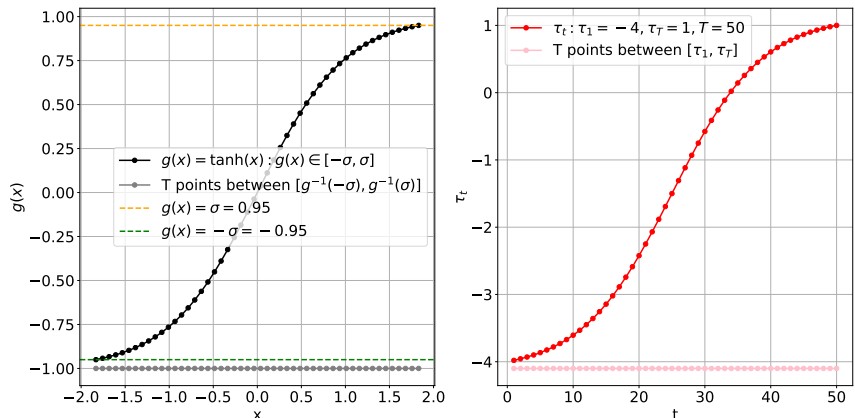

Figure 6: Illustration of Eq. (9) and visualization of our choice for $g(\cdot)$ and the resulted $\tau_t$ when $T = 50$. We use $g(\cdot) = \tanh(\cdot)$ (which can be other "S"-shape functions) and $\sigma = 0.95$ in our experiments. Here, we map the points on the black curve in the left plot to the points on the red curve in the right plot. Each gray point on the bottom of the left plot is from the $T$ evenly spaced x-coordinates between the x-interval $[g^{-1}(-\sigma), g^{-1}(\sigma)]$. We scale them to the $T$ t-coordinates in the bottom of the right plot (i.e., $t = 1, 2, \cdots, 50$), which associates with $T$ $\tau_t$ values represented by the red points between $[\tau_1, \tau_T]$ on the red curve.

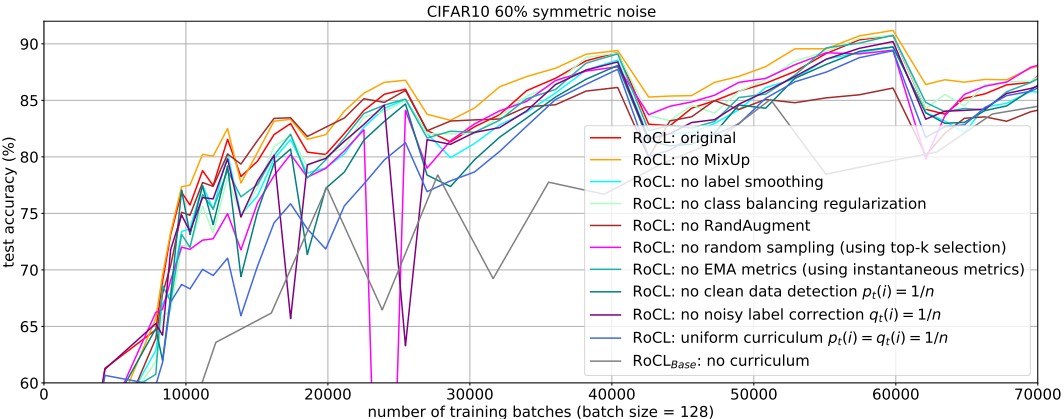

Figure 7: **Ablation study**: RoCL vs. its variants during the training of ResNet34 on **CIFAR10** containing **60% symmetric noises** on labels.

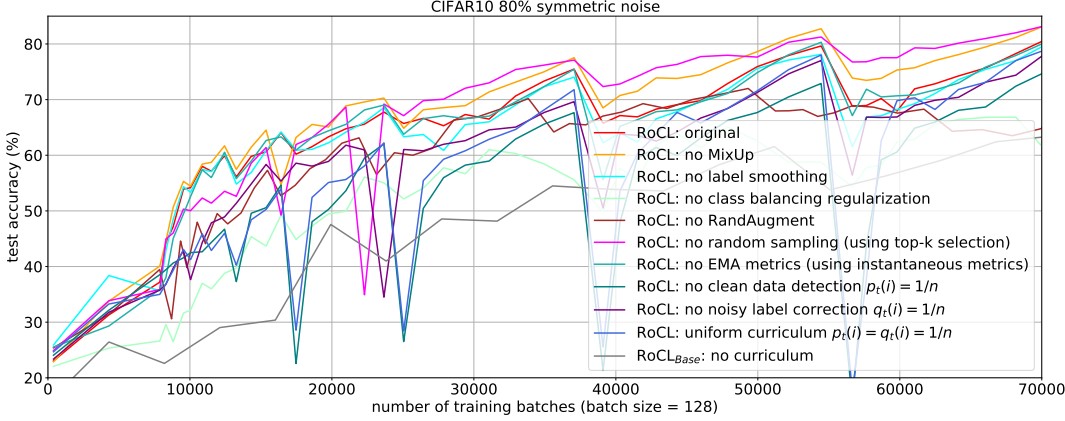

Figure 8: **Ablation study**: RoCL vs. its variants during the training of ResNet34 on **CIFAR10** containing **80% symmetric noises** on labels.

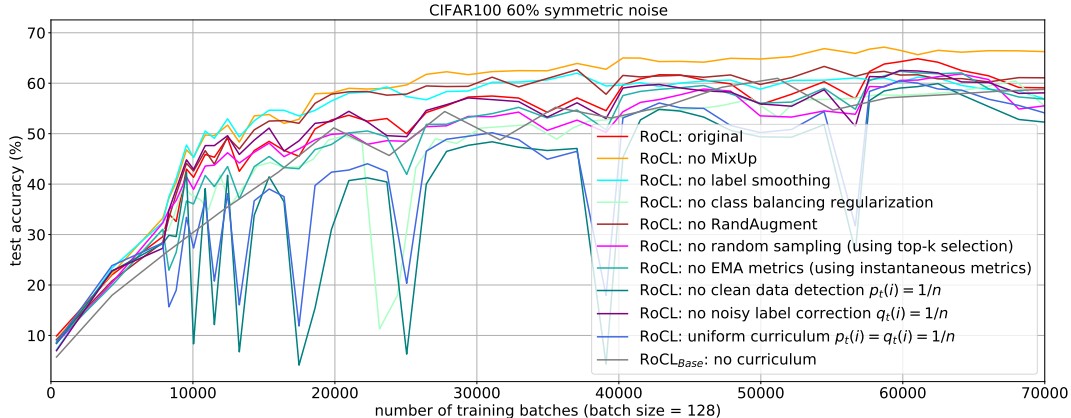

Figure 9: **Ablation study**: RoCL vs. its variants during the training of ResNet34 on **CIFAR100** containing **60% symmetric noises** on labels.

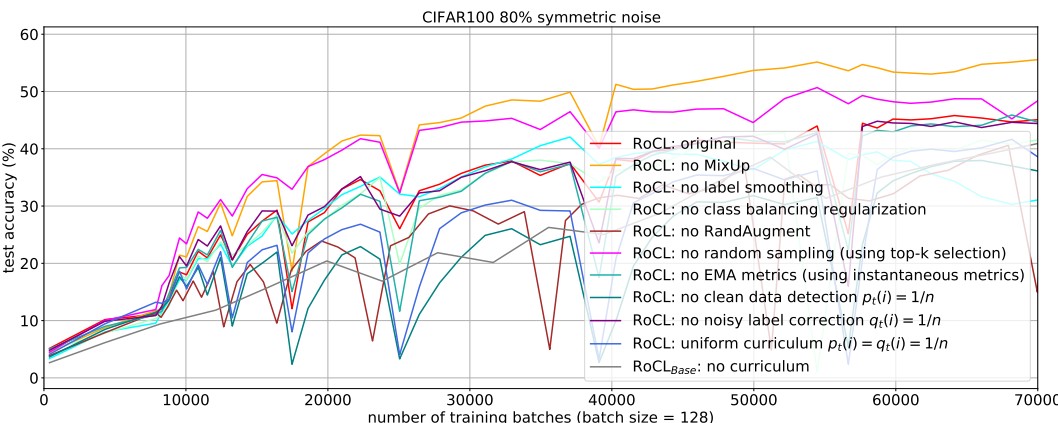

Figure 10: **Ablation study**: RoCL vs. its variants during the training of ResNet34 on **CIFAR100** containing **80% symmetric noises** on labels.

We present a more detailed analysis of the ablation study results with explanations of the observed phenomenons below.

- Most variants (except $RoCL_{Base}$, no RandAugment, and no ClassBalance) have similar performance as the original RoCL and perform better than or competitive with the SoTA results achieved by MentorMix. The differences compared to original RoCL become smaller under the lower noise rate setting (60%). $RoCL_{Base}$ uses all data for training in each step without applying any curriculum, showing that our proposed curriculum is the most critical component of RoCL in achieving the appealing improvements. Note $RoCL_{Base}$ already outperforms most methods in Table 3, which verifies the effectiveness of multi-episode training that alternates between supervised learning with the given labels and self-supervision with the pseudo labels.

- Removing RandAugment degrades the performance, especially when the noise rate is very high (e.g., 80%) because strong data augmentations are required by the self-supervision and the EMA consistency loss in RoCL, while trivial data augmentations can result in error accumulation or over-confidence in pseudo labels and inaccurate EMA consistency loss. The self-supervision aims to encourage the model output consistency over different augmentations of the same sample. Without augmentations with sufficient variations, self-supervision reduces to reinforcing the same outputs on similar samples and thus carries little information and can even magnify/accumulate errors (if any) in the original outputs. Also, the EMA consistency loss cannot generate meaningful consistency measures if computed on the same data or its trivial augmentations. Note a strong data augmentation is not always beneficial in all noisy label learning methods since it can increase the uncertainty in the presence of wrong labels, making the detection of clean data and noise correction more challenging. For example, we tried applying RandAugment to MentorMix (using

the official implementations of both) but observed inferior performance compared to the results using its original data augmentations.

- Class balance regularization is useful for the very high noise rate setting (80%), in which a wrong label may dominate the learning on a mini-batch by a large chance. However, when the noise rate is not that high (e.g., 60% on CIFAR10), removing it results in better performance.

- Although Mix-Up has been proved effective in previous methods, and for this reason, we followed MentorMix by starting with a relatively strong Mix-Up ($alpha = 8.0$) and then gradually reducing it to $\alpha = 0.2$. In the ablation study, we find that completely removing Mix-Up significantly improves performance. Mix-Up is helpful when applied to mix a clean label with a noisy label since the latter can be mediated with the former and thus softened. However, this is rarely the case for RoCL since RoCL either mainly learns from clean data or wrongly-labeled data with correct pseudo labels, and the transition between the two phases is short. When applied to two correct labels/pseudo labels, Mix-Up weakens each label's confidence, and we may lose information from the inter-class probabilities in the soft pseudo labels.

- Replacing weighted sampling with top-k selection ("no RandSampling") or replacing EMA metrics with instantaneous metrics ("no EMA metrics") causes less degeneration on the final test accuracies. However, they are important to the early-stage exploration and accurate estimation of EMA metrics on less-visited samples. In Figure 7-10, these two variants usually suffer from low accuracy and convergence speed during early stages. The only exception is "no RandSampling" in Figure 10, which performs better than the original RoCL. A possible reason is that the randomness brought by high uniform label noises already bring sufficient randomness for exploration.

- Replacing $p_t(i)$, $q_t(i)$ or both with uniform probabilities over all samples reduces the final test accuracies in all cases, e.g., the degradation is significant on CIFAR100 with 80% noise. In Figure 7-10, we can see that by setting $q_t(i) = 1/n$ results in less degradation than the other two. This is due to the more accurate pseudo labels generated for more data (even the ones with larger EMA consistency loss) as training proceeds. Moreover, since we are conservative in setting $\lambda_T$ and $\tau_T$, the performance is not very sensitive to wrong pseudo labels.

