# OpenReview forum: "Robust Curriculum Learning: from clean label detection to noisy label self-correction"
_ICLR.cc/2021/Conference — ICLR 2021 Poster_

### Official Review · AnonReviewer3 · 2020-10-27
**After the discussions I found that one of my comments was wrong and now I revised the review**

**Rating:** 6
**Confidence:** 5

**Review:**

This paper proposes a curriculum learning method to handle noisily labeled data. The idea is to introduce a consistency measure instead of directly apply a loss function for the typical supervised learning, where the specific consistency is measure for both temporal dimension along neighboring steps and spatial dimension over different data augmentation samples for a given real sample. The consistency measure is applied for self-supervision while the loss function is applied for supervised learning. The final optimization is managed between the two components through weighting parameters, such that the training is made through a migration from a supervised learning to self-supervision by gradually adapting the weighting parameters. Evaluations are reported on Cifar10/100, WebVision, and ILSVRC2012.

The motivation and rationale of this work makes sense, and the proposed method appears to be correct, at least at the conceptual level though I did not check in detail. After the discussions and the interactions with the authors, it came to my attention that one of my comments was wrong - I looked at a different paper that led me to have the conclusion that the authors cited a wrong performance number from a competitor in the literature; it turns out that I was wrong and I apologize to the authors.

The paper at the initial submission version read very rough, with a lot of grammatical errors, typos, or misleading/incorrect statements. In the experiment section, it is stated that three datasets were used for evaluations but in Table 1 it appears there is the fourth dataset ILSVRC2012 used, which never mentioned in the text, nor is mentioned in the paper on how that dataset is used (a portion like WebVision or the full). The whole paper ended up with no conclusion or discussion. After the discussions with the authors, it became clearer that the mentioned three datasets in the text were for training and ILSVRC2012 was used for evaluation. But still it would be a lot clearer to have such a statement in the text.

I had the comment that the title of the paper was misleading. The title reads: ROBUST CURRICULUM LEARNING: FROM CLEAN LAEL DETECTION TO NOISY LABEL SELF-CORRECTION. However, the proposed method, together with all the reported evaluations, focuses on learning the noise (in the labels); neither clean label detection nor noisy label correction is addressed. The authors disagreed with me but I was still not convinced by their argument. For noisy label self-correction, it may be relevant; but for clear label detection, I don’t think so.

I had a comment regarding the scale of the datasets used in the evaluations. But after having read the competitors’ work such as MentorMix, I now took it back and agreed with the authors. On the other hand, I agree with the comments raised by the other reviewers on lacking the ablation studies. I appreciated the authors’ efforts to report back the ablation studies, though only in part, and the results appeared to be convincing to me.

So overall, after the discussions and the revision provided by the authors, I am convinced that the paper is above the acceptance threshold. The paper does have presentation issue, and lacks extensive ablation studies.

---

> ### Author Response · Authors · 2020-11-14
> **Response to AnonReviewer3**
>
> Thanks for the comments! While we appreciate the time and effort you spent reviewing our paper, we believe that several of your comments are **incorrect** even though the paper explicitly clarifies these issues.
>
>
> **(1) "First, the title of the paper is misleading. The title reads: ROBUST CURRICULUM LEARNING: FROM CLEAN LAEL DETECTION TO NOISY LABEL SELF-CORRECTION...neither clean label detection nor noisy label correction is addressed."**
>
> The proposed curriculum (details in Section 3.1 and Algorithm 1) is exactly a smooth transition from (i) supervised learning using the detected clean labels as training targets to (ii) self-supervision using self-corrected pseudo-labels as training targets. One can also see this in Eq.7. In earlier epochs of each episode, we detect clean data as the samples with small EMA losses (large $p_t(i)$ in Eq.7) and train the model only using this clean data, when the supervised loss dominates the objective (large $\lambda$ in Eq.7). In later epochs, for samples with wrong labels (i.e., the ones with large EMA losses and thus small $p_t(i)$ in Eq.7) but consistent pseudo-labels (large $q_t(i)$ in Eq.7), we correct their wrong labels with the pseudo-labels by making the consistency loss dominate the objective (small $\lambda$ in Eq.7).
>
> **(2) Comparison on the first 50 classes of WebVision: "Third and more importantly, the reported evaluation comparison is NOT fair"**
>
> The comparison on the first 50 classes (called the "mini" subset in some papers) IS FAIR since the baseline accuracies are results on mini-WebVision and mini-ImageNet reported in the MentorMix paper, i.e., (Jiang et al., 2020). This mini-setting is a very common benchmark in the noisy-label learning literature. For example, **MentorMix's result is from the "Mini"-setting on the bottom of Table 4 from (Jiang et al., 2020) instead of the "Full"-setting**. Let me quote from (Jiang et al., 2020): "Following prior studies, we train our method on both the full training set (2.4M images on 1K classes) and the mini subset (61K images on 50 classes), and test it on two clean validation sets from ImageNet ILSVRC12 and WebVision." We followed **EXACTLY** the same setting of the mini subset in our paper submission.
>
> We explicitly state in two places, **first in the text on the left of Table 1, and also in caption of Table 1 (both on page 7 of our paper)**, that the reported validation accuracies are for models trained on the first 50 classes of WebVision. Thus, we hid nothing, and our comparison is fair.
>
> **(3) "...there is the fourth dataset ILSVRC2012 used, which never mentioned in the text.", "For ILSVRC2012, since there is no documentation at all in the paper, I have no idea whether the full set is used or only part of it...".**
>
> **We did not train any model on ILSVRC2012 or its mini-subset in this paper**. We only used its validation set to evaluate the model trained on the first 50 classes of WebVision. We explicitly stated it in the caption of Table 1 (page 7), quoted here: "Accuracy (\%) evaluated on WebVision and ILSVRC2012 validation sets for DNNs trained by noisy-label learning methods on WebVision training set (first 50 classes)". We will, however, clarify in the next version of the paper that WebVision and ILSVRC2012 have the same class labels to avoid any confusion. Thanks for pointing this out.
>
> **(4) "The caption in Fig. 1 is not consistent with the description in text."**
>
> Sorry, can you please specify which part of the text is not consistent with the caption of Fig.1? We are happy to clarify/explain it. Note that the loss plots in Fig.1 are discussed at the bottom of Section 2.1 while the consistency plots in Fig.1 are discussed at the bottom of Section 2.2.

---

> > ### Comment · AnonReviewer3 · 2020-11-20
> > **Response to authors' response to my comments**
> >
> > I would like to ask the authors to read my comments carefully before responding them. That would have saved all of our time. Let me briefly answer your response.
> > (1) I understand that this work was about curriculum learning. Please keep in mind - clean label detection is a different problem; similarly, though curriculum learning can be arguably related to noisy label self-correction, the latter can be also considered as a different problem. Clearly, this work has nothing to do with either of them. That's why I said the title was misleading.
> > (2) I understood that you followed the same setting of the mini subset in the experiments. What I meant was that the performance data you quoted and reported from the baselines in the paper were actually obtained for the full dataset, NOT for the mini subset. That is NOT fair. If you used the correct data from the baselines, it would be a completely different story!
> > (3) Let me quote what you stated in your paper (beginning of Experiments section): "We evaluate RoCL with other state-of-the-art approaches for noisy-label learning on three widely used benchmarks, i.e., CIFAR10/100 with two types of synthetic noises (i.e., symmetric and asymmetric), and WebVision (Li et al., 2017a) (the first 50 classes) containing unknown noises from web labels. " On the other hand, you reported data for ILSVRC2012 in Table 1. Was I right in my comments?

---

> > > ### Author Response · Authors · 2020-11-20
> > > **Response to AnonReviewer3's Response**
> > >
> > > Thanks for the response! But we believe that the three points in your response are **all incorrect**:
> > >
> > > **(1) I understand that this work was about curriculum learning. Please keep in mind - clean label detection is a different problem; similarly, though curriculum learning can be arguably related to noisy label self-correction, the latter can be also considered as a different problem. Clearly, this work has nothing to do with either of them. That's why I said the title was misleading**
> > >
> > > --In our method:
> > >
> > > **Clean data selection**: we detect clean data as the samples with small EMA loss. Since we detect them as clean data, we use their given labels for supervised learning.
> > >
> > > **Noisy label correction**: we correct the noisy labels by pseudo labels for samples with large EMA loss (indicating the given label is wrong) but small EMA consistency loss (indicating the pseudo label is correct). We use the corrected labels for self-supervision on these samples.
> > >
> > > **(2) I understood that you followed the same setting of the mini subset in the experiments. What I meant was that the performance data you quoted and reported from the baselines in the paper were actually obtained for the full dataset, NOT for the mini subset. That is NOT fair. If you used the correct data from the baselines, it would be a completely different story!**
> > >
> > > -- The performance data we quoted from the baseline papers in Table 1 were obtained for **the mini-setting as stated in their papers, so the comparison IS FAIR!** Can you specify which exact numbers in Table 1 are for the full dataset? Note we specified the exact location of a previous paper where Table 1's results come from: "For example, MentorMix's result is from the "Mini"-setting on the bottom of Table 4 from (Jiang et al., 2020) instead of the "Full"-setting." If you do not believe it, can you spend one minute to check **Table 4 in MentorMix paper**? Let me quote the last row of Table 4 here: **"Mini Ours (MentorMix) 72.9(91.1) 76.0(90.2)"**, which are the same numbers in our Table 1 for MentorMix.
> > >
> > > **(3) Let me quote what you stated in your paper (beginning of Experiments section): "We evaluate RoCL with other state-of-the-art approaches for noisy-label learning on three widely used benchmarks, i.e., CIFAR10/100 with two types of synthetic noises (i.e., symmetric and asymmetric), and WebVision (Li et al., 2017a) (the first 50 classes) containing unknown noises from web labels. " On the other hand, you reported data for ILSVRC2012 in Table 1. Was I right in my comments?**
> > >
> > > -- As clearly stated in Table 1's caption, and let me emphasize here: **ILSVRC2012 is ONLY used for VALIDATION, not LEARNING**, since it does not contain any label noise! Let me re-quote what you quoted from our paper above: "...for noisy-label **LEARNING** on three widely used benchmarks"---ILSVRC2012 is not among the three benchmarks here because **no noisy label LEARNING happens on ILSVRC2012!** In Table 1, we only use it as a **VALIDATION** set for the model achieved by noisy label learning on mini-WebVision only (which does not include ILSVRC2012).

---

### Official Review · AnonReviewer1 · 2020-10-27
**Interesting work on Robust Curriculum Learning via interplay between loss and consistency**

**Rating:** 5
**Confidence:** 4

**Review:**

#Summary

This paper proposes a robust curriculum learning method that interpolates a regular loss and a consistency loss, aiming at a smooth transition from learning from clean data and then to noisy data with pseudo labels.

#Pros
- The paper is well-written, and the results over several benchmark datasets seem to be strong.
- Some of the insights provided in this paper, seem rather interesting, like by transitioning from supervised learning, to self-supervised learning of noisy data, can better benefit the learning process.

#Cons
- The authors should perform an ablation study of the RoCL method. Currently the final proposed method mixes too many components, and it is hard to disentangle the true contribution of each component. For example, in section 3.4, the authors mentioned additional techniques were added, like class-balance regularization, label-smoothing, and mix-up, a further analysis is required to understand the true contribution for each individual part.

- Similarly, for all the baselines used, the authors should do a better categorization of each baseline method, to ensure a fairer comparison. For example, does any of the baselines use model averaging, mix-up, label-smoothing, or data augmentations? How is RoCL without mix-up compared to baselines that don't use mix-up?

- If I understand correctly, one of the key contributions is the interplay between the regular loss and the consistency loss, but the scheduling part is not super principled and seems to involve a lot of ad-hoc tuning of the balancing parameter $\lambda$ and the temperatures. Is there a principled way to balance the two losses?

- The RoCL algorithm seems to involve a lot of parameters. The averaging parameter $\gamma$ for EMA, $\lambda$ for the trade-off between loss and consistency, and tempature $\tau_1, \tau_2$ (and the additional params required for scheduling the temperatures properly). In practice, how are those parameters picked? Is there a lot of careful tuning required?

- The paper seems to have combined a lot of existing techniques. The paper would be stronger if the authors can provide further analysis to better understand how/why RoCL works. E.g., how important is EMA/data augmentation/data sampling respectively?


#Overall recommendation

Overall I'm on the fence but tend to reject. I think this paper can benefit a lot from better organizing of the methods and results, with a clearer focus on its major contributions. Currently the experimental result is a mixture of multiple existing approaches and the proposed RoCL method, it is hard to know what the role of each approach is and whether the proposed RoCL method indeed improves curriculum learning. I think this paper has the potential of providing some great insights, but the current set of results are rather noisy.

#Minor comments
- Figure 1 is really small and hard to read.
- The paper needs to be better organized. Currently it seems like the authors run out of space and rushed through the experimental results.
- Can you clarify if the $\gamma$ is the same for Eq. 1, Eq. 3, and for computing $\bar{\theta}_t$? If they are different please use different notations.
- I don't quite get Eq. 9, why is the temperature defined in this way?
- How important is the sampling part in Algorithm 1? It involves another parameter $b_k$ and how sensitive is RoCL to the choice of that parameter?
- The consistency loss over augmented examples is also a commonly-adopted technique in semi-supervised learning, citations to the use of those methods in existing literature are missing, e.g., [1] Mehdi Sajjadi, Mehran Javanmardi, and Tolga Tasdizen. Regularization with stochastic transformations and perturbations for deep semi-supervised learning. NeuIPS 2016.
[2] Samuli Laine and Timo Aila. Temporal ensembling for semisupervised learning. ICLR 2017.

---

> ### Author Response · Authors · 2020-11-23
> **Response to AnonReviewer1: Part I**
>
> Thanks for your comments! We have added a complete ablation study in the new version of the paper including most experiments suggested in your comments. Here are our detailed replies to your questions. We have modified the paper according to your comments and added citations to the suggested papers.
>
> **(1) The authors should perform an ablation study of the RoCL method. The paper would be stronger if the authors can provide further analysis to better understand how/why RoCL works. E.g., how important is EMA/data augmentation/data sampling respectively?**
>
> In the new version, we provide a thorough ablation study including 10 variants of RoCL and discuss the importance of the applied techniques in RoCL. They include variants without using class-balance regularization, label-smoothing, mix-up, EMA metrics, data augmentation and data sampling.
>
> **(2) Does any of the baselines use model averaging, mix-up, label-smoothing, or data augmentations? How is RoCL without mix-up compared to baselines that don't use mix-up?**
>
> In the revision, we added marks to the baselines in the Tables to report which techniques have been used in these baselines. In our new ablation study, we found that removing mixup from RoCL can actually bring significant improvements on some experiments. Although mixup can make samples with wrong labels less harmful by mixing it with other clean data when the training batches are uniformly sampled, it is less useful in RoCL since the samples we select are either associated with correct labels or correct pseudo labels and wrong labels are rarely used for training. Moreover, mixup can be harmful in our self-supervision stage since the pseudo labels are soft class probabilities produced by the model itself, which can be close to uniform distribution but still informative. However, applying mixup to two groups of soft probabilities may make the mixed pseudo labels even closer to uniform and wipe out the useful inter-class correlation in the original soft class probabilities.
>
> **(3) But the scheduling part is not super principled and seems to involve a lot of ad-hoc tuning of the balancing parameter and the temperatures. Is there a principled way to balance the two losses? In practice, how are those parameters picked? Is there a lot of careful tuning required?**
>
> In summary, $\lambda$ controls the transition between supervised learning loss and self-supervised learning loss, while $\tau_1$ controls the selection of data with correct/wrong labels and $\tau_2$ controls the selection of data with correct/wrong pseudo-labels. We did not heavily tune the hyper-parameters $\lambda$ and the temperatures $\tau_{1,2}$ but only followed a principle and tried a limited number of choices in experiments. We do not need to carefully tune them since they are not static but gradually changing in our curriculum, and thus the performance is not very sensitive to their starting and ending values. We apply one interpolation function to generate the schedules for three of them. We do have a principle to set up the starting and ending values for them, as stated in Section 3.1 (mainly in the last paragraph), which is the transition from supervised learning on clean data to self-supervised learning on noisy data, and here is the summarization:
>
> (i) For $\lambda$, we start from a value close to 1 and end with a value close to 0, because our curriculum is a transition from supervised learning ($\lambda=1$) to self-supervised learning ($\lambda=0$). Its schedule is coupled with that of $\tau_1$ as illustrated below. We do not balance the two losses since they should be applied mainly to different data in different phases. In fact, we keep only one of them dominating the objective at most times and make the transition between them short (please see (ii) below for details).
>
> (ii) For $\tau_1$ in Eq. (5)-(6), we start from a negative value and end with a positive value since we select clean data for early supervised learning phase and gradually transition to a self-supervision phase on wrongly-labeled samples, and negative $\tau_1$ produces large probabilities $p_t(i)$ for clean data, while positive $\tau_1$ produces large probabilities $p_t(i)$ for wrongly-labeled data. We apply an "s"-shape function for interpolation between the two values so for most of time either supervised learning or self-supervised learning dominates and the transition phase between the two (with more uncertainty and for exploration) is short. This gives us a schedule $\tau_{1:T}$ for $\tau_1$.

---

> > ### Author Response · Authors · 2020-11-23
> > **Response to AnonReviewer1: Part II**
> >
> > (iii) For the schedule of $\tau_2$ in Eq. (5)-(6), we use the reversed sequence $\tau_{T:1}$ from $\tau_1$'s schedule, i.e., we start from a negative $\tau_2$ and end with a positive $\tau_2$. Positive $\tau_2$ produces large probabilities $q_t(i)$ for data with wrong pseudo-label. Together with a positive $\tau_1$, they aim to select clean data that the model does not fully learn during the supervised learning phase. On the other hand, negative $\tau_2$ together with positive $\tau_1$ in the self-supervised learning phase aim to select wrongly-labeled data with correct pseudo-labels.
> >
> > (iv) We set their exact values based on observations in Figure 1: clean data detection is easier but the detection of correct pseudo-labels is harder. Hence, we can be more confident on the clean data detection and thus we set the starting value $\tau_1$ to be large in magnitude, but we need to be more conservative about the ending value $\tau_T$ and keep it closer to 0. In experiments, we tried $\tau_1=\{-4,-3\}$ and $\tau_T=\{1,2\}$ and finally chose $\tau_1=-4$ and $\tau_T=1$ since this choice performs consistently well on all experiments, though it might not be the best choice for all. We did not try larger values for them since we need a certain amount of exploration but increasing their magnitudes quickly degenerate the weighted sampling to top-k selection. For the similar reason from Figure 1, we set the starting value $\lambda_1$ to be closer to 1 than the ending value $\lambda_T$ to 0. In experiments, we set $\lambda_1=0.9$ and did not try other values for it, we tried $\lambda_T=\{0.1,0.2,0.3\}$ and on some experiments the first two values lead to performance degradation.
> >
> > **(4) Questions about discounting factor $\gamma$ in EMA.**
> >
> > $1-\gamma$ is the discounting factor in exponential moving average and it is commonly set as a value close to 1 (i.e., $\gamma$ close to 0). In this paper, we simply set $\gamma=0.1$ and it works well on all experiments. We tried $0.05$ and $0.15$ for them but the resulted performance stays almost the same.
> >
> > We use the same $\gamma$ for Eq. (1), Eq. (3) and $\bar \theta_t$. We also clarified it in the new version of the paper.
> >
> > **(5) I don't quite get Eq. 9, why is the temperature defined in this way?**
> >
> > Eq. 9 is a schedule for the temperature parameters given by the interpolation between $\tau_1$ and $\tau_T$ using an "s"-shape function $g(\cdot)$. In the revision, we modified the equation and its explanations. We also illustrate the equation graphically in Figure 6 of the Appendix. It is essentially an affine mapping from the points on an "s"-shape function like $\tanh$ to the interval $[\tau_1, \tau_T]$. Please see the reason for using an "s"-shape function below Eq. (9) and explained in the above reply (ii) of the question (3).
> >
> > **(6) How important is the sampling part in Algorithm 1? It involves another parameter $b_k$ and how sensitive is RoCL to the choice of that parameter?**
> >
> > The sampling part in Algorithm 1 is our proposed curriculum. The algorithm reduces to the Baseline algorithm in Algorithm 2 if removing the sub-sampling and using all samples for training in each step. And the performance becomes much worse for the Baseline. $b_k$ is the number of samples selected for training in episode-$k$ and we gradually increase it in line-16 of Algorithm 1. This schedule has been widely used in previous curriculum learning methods (Benjio 2009, Kumar et al., 2010, Jiang et al. 2015, Zhou & Bilmes 2018). In our experiments, we start from 20% of the whole training set and gradually increase the budget by a multiplicative factor 1.1 every episode. This works consistently well on all experiments so we did not try other choices for it.

---

### Official Review · AnonReviewer2 · 2020-10-27
**This article focuses on model training tasks under noisy labels. The authors combine classification loss and prediction inconsistency and complete the shift the training strategies from supervision to self-supervision with the hyperparameters' schedule.**

**Rating:** 6
**Confidence:** 5

**Review:**

This article is concerned with the problem of training models under noisy data. The authors first adopt the loss and output consistency for data selection. EMA method is used for smoothing to obtain more accurate clean label detection. Meanwhile, through the introduction of temperature hyperparameters, the model gradually completes the transition from supervised learning using clean labels to self-supervised learning using noisy labels.

Strength:
1.	The authors use both loss function and prediction invariance for sample selection. The proposed model adjusts the sample selection strategy for different training stages to obtain a more informative training sample.
2.	The authors used the EMA algorithm to smooth the sample selection metrics, resulting in better clean sample detection performance.
3.	The change of hyperparameters allows the model to accomplish the change of sample selection strategy and transform the training strategy of the model from fully supervised to self-supervised.
4.	The authors provide a detailed theoretical analysis and experimental demonstration of the proposed method, which achieves SOTA performance in both the CIFAR10/100 and WebVision datasets

Weakness:
1.	As several modifications mentioned in Section 3.4 were used, it would be better to provide some ablation experiments of these tricks to validate the model performance further.
2.	The model involves many hyperparameters. Thus, the selection of the hyperparameters in the paper needs further explanation.
3.	A brief conclusion of the article and a summary of this paper's contributions need to be provided.
4.     Approaches that leveraging noisy label noise label regularization and multi-label co-regularization were not reviewed or compared in this paper.

---

> ### Author Response · Authors · 2020-11-23
> **Response to AnonReviewer2: Part I**
>
> Thanks for your comments! We have added a complete ablation study in the new version of the paper including most experiments suggested in your comments. Here are our detailed replies to your questions.
>
>
> **(1) As several modifications mentioned in Section 3.4 were used, it would be better to provide some ablation experiments of these tricks to validate the model performance further.**
>
> In the new version, we provide a thorough ablation study including 10 variants of RoCL and discuss the importance of the applied techniques in RoCL.
>
> **(2) The model involves many hyper-parameters. Thus, the selection of the hyper-parameters in the paper needs further explanation.**
>
> We did not heavily tune the hyper-parameters but only followed a principle and tried a limited number of choices in our experiments. We do not need to carefully tune them since they are not static but gradually changing in our curriculum, and thus the performance is not very sensitive to their starting and ending values. We apply one interpolation function to generate the schedules for three of them. We do have a principle to set up the starting and ending values for them, as stated in Section 3.1 (mainly in the last paragraph), which is the transition from supervised learning on clean data to self-supervised learning on noisy data, and here is the summarization:
>
> (i) For $\lambda$, we start from a value close to 1 and end with a value close to 0, because our curriculum is a transition from supervised learning ($\lambda=1$) to self-supervised learning ($\lambda=0$). Its schedule is coupled with that of $\tau_1$ as illustrated below. We do not balance the two losses since they should be applied mainly to different data in different phases. In fact, we keep only one of them dominating the objective at most times and make the transition between them short (please see (ii) below for details).
>
> (ii) For $\tau_1$ in Eq. (5)-(6), we start from a negative value and end with a positive value since we select clean data for early supervised learning phase and gradually transition to a self-supervision phase on wrongly-labeled samples, and negative $\tau_1$ produces large probabilities $p_t(i)$ for clean data, while positive $\tau_1$ produces large probabilities $p_t(i)$ for wrongly-labeled data. We apply an "s"-shape function for interpolation between the two values so for most of time either supervised learning or self-supervised learning dominates and the transition phase between the two (with more uncertainty and for exploration) is short. This gives us a schedule $\tau_{1:T}$ for $\tau_1$.
>
> (iii) For the schedule of $\tau_2$ in Eq. (5)-(6), we use the reversed sequence $\tau_{T:1}$ from $\tau_1$'s schedule, i.e., we start from a negative $\tau_2$ and end with a positive $\tau_2$. Positive $\tau_2$ produces large probabilities $q_t(i)$ for data with wrong pseudo-label. Together with positive $\tau_1$, they aim to select clean data that the model does not fully learn during the supervised learning phase. On the other hand, negative $\tau_2$ together with positive $\tau_1$ in the self-supervised learning phase aim to select wrongly-labeled data with correct pseudo-labels.
>
> (iv) We set their exact values based on observations in Figure 1: clean data detection is easier but the detection of correct pseudo-labels is harder. Hence, we can be more confident on the clean data detection and thus we set the starting value $\tau_1$ to be large in magnitude, but we need to be more conservative about the ending value $\tau_T$ and keep it closer to 0. In experiments, we tried $\tau_1={-4,-3}$ and $\tau_T={1,2}$ and finally chose $\tau_1=-4$ and $\tau_T=1$ since this choice performs consistently well on all experiments, though it might not be the best choice for all. We did not try larger values for them since we need certain amount of exploration but increasing their magnitudes quickly degenerate the weighted sampling to top-k selection. For the similar reason from Figure 1, we set the starting value $\lambda_1$ to be closer to 1 than the ending value $\lambda_T$ to 0. In experiments, we set $\lambda_1=0.9$ and did not try other values for it, we tried $\lambda_T={0.1,0.2,0.3}$ and on some experiments the first two values lead to performance degradation.
>
> (v) $1-\gamma$ is the discounting factor in exponential moving average and it is commonly set as a value close to 1 (i.e., $\gamma$ close to 0). In this paper, we simply set $\gamma=0.1$ and it works well on all experiments. We tried $0.05$ and $0.15$ for them but the resulted performance stays almost the same.

---

> > ### Author Response · Authors · 2020-11-23
> > **Response to AnonReviewer2: Part II**
> >
> > **(3) A brief conclusion of the article and a summary of this paper's contributions need to be provided.**
> >
> > We added a conclusion session in the revision.
> >
> > **(4) Approaches that leveraging noisy label noise label regularization and multi-label co-regularization were not reviewed or compared in this paper.**
> >
> > In the new version of the paper, We added discussion of [Hu et al. (2019)] in the related works. Please let us know if you have suggestions about other works using label regularization related to noisy label training, and we are happy to add them for discussions.

---

### Official Review · AnonReviewer4 · 2020-10-28
**Interesting method, with many experiment and good performance, but some details missing**

**Rating:** 5
**Confidence:** 4

**Review:**

# Summary
The paper proposes a joint pseudo-labelling and curriculum learning strategy. It addresses the problem of training with noisy labels, especially robustly correcting noisy labels.

# Recommendation
Borderline paper, with a lot of strong points, as given below. My major concern, is that the improvement is e.g. due to more advanced data augmentation (Cubuk et al. (2020)). I think it should be easy for the authors to provide more details here, i.e. an ablation study, that could help in deciding. So far I am unfortunately not conviced, that the progress is due to the proposed method alone.

# Strong/Weak points
## Pros
Coupling the acquisition of pseudo-labels and selection of clean labels and especially a smooth curriculum using both is an interesting idea.
The proposed usage of the exponential moving average is reasonably motivated by the oscillating patterns of the instantaneous loss values.
The proposed algorithm is well embedded in recent publications and well motivated.
Multiple noise rates and multiple datasets are considered, proving the methods applicability.

## Cons
The origin of Equation 6 is not as clear as it should be. The description of the terms in eq. 6 could be improved, especially the fact, of "abuse notation".
While the coupling of $p_t(i)$ and $q_t(i)$ seems to be a good idea, the theoretical justification is not as convincing to me. Further theoretical background or experimental verification would be good to support the claim of this coupling being meaningful.
This and the fact, that a ablation study is missing makes it hard to judge the methods contribution.

# Questions to the authors
A more detailed comparison e.g. experimental between using $p_t(i)$ and $q_t(i)$ separatly, vs. using the jointly would be very interesting.

What data augmentation was used for all other methods, except RoCL? If all other results are taken from other papers, the aim should be to excatly reproduce the setting used there, or at least prove via an ablation study, the impact of each part of your training.

What happens if you do not use data augmentation at all?

Could you provide more details on $\tau_{1,2}$ and $\lambda$, so far do not see a justification for their respective values. Simple hyperparameter search? Is there a theoretical interpretation, or limits you could derive?

# Detailed comments
Eq. 2 is referenced before it is stated, consider rearranging.

Eq. 6 please clarify the description, e.g. what is "abuse notation", why are you replacing by instantaneous counterparts here?

Minor comment, the text in Figure 1 could be larger


Minor typo "Simply removing noisy data from training discards important information about data distribution." --> "the data distribution", some more typos in that same paragraph, please take some time to correct them.

---

> ### Author Response · Authors · 2020-11-23
> **Response to AnonReviewer4: Part I**
>
>
> Thanks for your comments! We have added a complete ablation study in the new version of the paper including most experiments suggested in your comments. Here are our detailed replies to your questions.
>
> **(1) My major concern, is that the improvement is e.g. due to more advanced data augmentation (Cubuk et al. (2020)). What happens if you do not use data augmentation at all?**
>
> **Data augmentation plays a key role in the self-supervision/self-correction stage of RoCL**: stronger data augmentations can encourage model consistency within a larger local region around each selected sample, while self-supervision with weak data augmentation can be harmful to noisy label learning. Without data augmentation (or with only trivial augmentations such as random horizontal flips), the consistency-loss driven self-supervision is much less useful and can even be harmful when noise rates are high. In such a case, self-supervision utilizes the output of the same input (or very similar inputs in the case of trivial data augmentations) as its training target (ref. Eq. 2), which could magnify or accumulate errors (if any) from the network output. Moreover, to get an EMA consistency metric that more precisely reflects the correctness of the pseudo-labels (so we can avoid selecting wrong pseudo-labels), we need to evaluate the consistency of model outputs over more variations of each sample. Without strong data augmentations, it is likely to get wrong pseudo-labels, which can be harmful in the training process.
>
> Hence, you are right that the data augmentations are important to the success of RoCL because (1) data augmentations are critical for self-supervision, and the self-supervision training is an essential component of our curriculum;
> (2) the quality of EMA consistency loss also highly depends on the variations of samples. Although data augmentation is important, our method does not solely depend on data augmentation: for example, applying self-supervision to samples with wrong pseudo-labels is definitely harmful, and our curriculum is capable in selecting correct pseudo-labels for self-supervision using the EMA consistency loss.
>
> In the newly added ablation study, we only apply the trivial horizontal flip and random crop for data augmentation, same in many previous methods. The results show that the test accuracy decreases in later episodes of RoCL when the noise rate is high (80%) and this is mainly caused by the error accumulation of self-supervision and inaccurate EMA metrics under weak data augmentation.
>
> **We also tried applying RandAugment (Cubuk et al. (2020)) to one of the best baseline methods, MentorMix** (we used Google's official implementation for RandAugment and MentorMix), and observed inferior performance compared to the results using its original data augmentations, as reported in Table 5. We conjecture that the large variations of RandAugment may be harmful to the mentor part of MentorMix to generate accurate rating of samples, while our method can benefit from RandAugment due to our curriculum to select samples with accurate pseudo-labels.
>
> **(2) ...provide more details here, i.e. an ablation study, that could help in deciding. So far I am unfortunately not convinced, that the progress is due to the proposed method alone.**
>
> We have added a thorough ablation study in the updated version. Note that straightforward evidence is already available in the original version, which is the comparison between the Baseline (Algorithm 2 in the Appendix) and RoCL. The Baseline algorithm uses all the techniques as RoCL, including MixUp, the same data augmentations, label smoothing, etc. The only difference is that RoCL selects a subset of samples for each epoch using the proposed curriculum, while the Baseline simply uses all the training samples. Even with all the existing techniques applied, without our newly proposed curriculum, one can observe a large performance gap between them in Figure 2-5 in the Appendix, which demonstrates the effectiveness of the proposed curriculum.
>
> Some of the techniques are tightly coupled with our curriculum, i.e., applying these techniques uniformly to all available samples without our curriculum is inefficient or even harmful to the training process. For example, when using data augmentations for the consistency loss, if we apply the augmentations uniformly to all samples, we may end up selecting samples with wrong pseudo-labels computed based on the augmentations, which can decrease the model performance, and we also waste the computations on augmenting these samples.
>
> **(3) A more detailed comparison e.g. experimental between using $p_t(i)$ and $q_t(i)$ separately, vs. using the jointly would be very interesting.**
>
> In the newly added ablation study, we compare RoCL with three variants: (1) using $p_t(i)$ only while $q_t(i)$ always set to be uniform; (2) using $q_t(i)$ only while $p_t(i)$ always set to be uniform; (3) both $p_t(i)$ and $q_t(i)$ are always set to be uniform.

---

> > ### Author Response · Authors · 2020-11-23
> > **Response to AnonReviewer4: Part II**
> >
> > **(4) Could you provide more details on $\tau_{1,2}$ and $\lambda$, so far do not see a justification for their respective values. Simple hyper-parameter search? Is there a theoretical interpretation, or limits you could derive?**
> >
> > We did not heavily tune the hyper-parameters but only followed a principle and tried a limited number of choices in experiments. We do not need to carefully tune them since they are not static but gradually changing in our curriculum, and thus the performance is not very sensitive to their starting and ending values. We apply one interpolation function to generate the schedules for three of them. We do have a principle to set up the starting and ending values for them, as stated in Section 3.1 (mainly in the last paragraph), which is the transition from supervised learning on clean data to self-supervised learning on noisy data, and here is the summarization:
> >
> > (i) For $\lambda$, we start from a value close to 1 and end with a value close to 0, because our curriculum is a transition from supervised learning ($\lambda=1$) to self-supervised learning ($\lambda=0$). Its schedule is coupled with that of $\tau_1$ as illustrated below.
> >
> > (ii) For $\tau_1$ in Eq. (5)-(6), we start from a negative value and end with a positive value since we select clean data for early supervised learning phase and gradually transition to a self-supervision phase on wrongly-labeled samples, and negative $\tau_1$ produces large probabilities $p_t(i)$ for clean data, while positive $\tau_1$ produces large probabilities $p_t(i)$ for wrongly-labeled data. We apply an "s"-shape function for interpolation between the two values so for most of time either supervised learning or self-supervised learning dominates and the transition phase between the two (with more uncertainty and for exploration) is short. This gives us a schedule $\tau_{1:T}$ for $\tau_1$.
> >
> > (iii) For the schedule of $\tau_2$ in Eq. (5)-(6), we use the reversed sequence $\tau_{T:1}$ from $\tau_1$'s schedule, i.e., we start from a negative $\tau_2$ and end with a positive $\tau_2$. Positive $\tau_2$ produces large probabilities $q_t(i)$ for data with wrong pseudo-label. Together with positive $\tau_1$, they aim to select clean data that the model does not fully learn during the supervised learning phase. On the other hand, negative $\tau_2$ together with positive $\tau_1$ in the self-supervised learning phase aim to select wrongly-labeled data with correct pseudo-labels.
> > (iv) We set their exact values based on observations in Figure 1: clean data detection is easier but the detection of correct pseudo-labels is harder. Hence, we can be more confident on the clean data detection and thus we set the starting value $\tau_1$ to be large in magnitude, but we need to be more conservative about the ending value $\tau_T$ and keep it closer to 0. In experiments, we tried $\tau_1=\{-4,-3\}$ and $\tau_T=\{1,2\}$ and finally chose $\tau_1=-4$ and $\tau_T=1$ since this choice performs consistently well on all experiments, though it might not be the best choice for all. We did not try larger values for them since we need a certain amount of exploration but increasing their magnitudes quickly degenerate the weighted sampling to top-k selection. For a similar reason from Figure 1, we set the starting value $\lambda_1$ to be closer to 1 than the ending value $\lambda_T$ to 0. In experiments, we set $\lambda_1=0.9$ and did not try other values for it, we tried $\lambda_T=\{0.1,0.2,0.3\}$ and on some experiments the first two values lead to performance degradation.
> >
> >
> > **(5) Detailed comments**
> >
> > We have modified the paper accordingly. The "abuse of notation" refers to $q_t(i), p_t(i), \mathcal P(i)$ in Eq. (7)-(8) since they are computed from the instantaneous metrics in these two equations but their definitions in Eq.(4)-(5) use EMA metrics. In the revision, we removed the poor notation by using different notation $q'_t(i), p'_t(i), \mathcal P'(i)$ for Eq. (7)-(8), to distinguish from the definition in Eq.(4)-(5).

---

> > > ### Comment · AnonReviewer4 · 2020-11-25
> > > **Post-Rebuttal**
> > >
> > > I closely read the other reviewers comments and the authors response. I would like to thank the authors for the additional details provided. The additional ablations provide interesting insights.
> > >
> > > As pointed out by the other reviewers as well it remains still unclear, which of the many concepts applied contributes to what extend to the success of RoCL.  I detailed my impression for data augmentation below and still see that as the main concern with regards to evaluating the efficacy of the proposed RoCL approach. However parts of it are still more gut feeling than hard conclusion from the provided experiments. While I feel the ablation study is going in a good direction, still some parts are missing, some suggestions see also below.
> > >
> > > From my understanding Mix-Up hinders the performance of RoCL, hence standard RoCL should probably be defined without Mix-up.
> > >
> > > Mentor-Mix uses "standard data augmentation" (quoted from the Mentor-Mix paper), while RoCL uses Rand-Augment, Rand-Augment alone increases the accuracy on CIFAR10 around 1%, as reported by Cubuk et al. (Note they use WRN, not directly transferable to the ResNet34 used for RoCL), with RandAugment being a much stronger augmentation.
> > >
> > > From Table 5 I conclude, **if** RoCL and Mentor-Mix use **the same data augmentation** Mentor-Mix outperforms RoCL on CIFAR10 and the other way round on CIFAR100. Further it might be the case that Mix-Up and stronger Augmentation (RandAugment) play not well together, hence Mentor-Net (w/o Mix-up) + Rand-Augment would be the direct comparison to the original RoCL.

---

> > > > ### Author Response · Authors · 2020-11-25
> > > > **Response to AnonReviewer4's new response**
> > > >
> > > > Hi AnonReviewer4,
> > > >
> > > > Thanks for your response! **According to your new response, we now start to run MentorMix experiments w/o MixUp but using RandAugment. However, it might be too late since the deadline for discussion has already passed.** At the same time, we would like to emphasize that RoCL significantly outperforms MentorMix when both methods use RandAugment. Moreover, RoCL achieves new SoTA performance and significantly outperforms previously reported results on almost all benchmarks presented in the paper.
> > > >
> > > > We argue that **although RandAugment improves RoCL, it may not be the same case for methods that do not use self-supervision (e.g., MentorMix), and here is a brief explanation.**
> > > >
> > > > RandAugment improves RoCL, mainly because that the self-supervision plays an important role in RoCL, and self-supervision needs strong data augmentations to stably work under label noise. **However, this is not the same case for MentorMix and other methods that do not use self-supervision**. Hence, the improvement (if any) might be very limited for them.
> > > >
> > > > In fact, strong data augmentation brings more uncertainty in addition to the label noise, so it does not always bring improvement to noisy label learning methods. This is a critical difference from the clean data scenario studied in the RandAugment paper.
> > > >
> > > > Quote from your new comments: **"if RoCL and Mentor-Mix use the same data augmentation Mentor-Mix outperforms RoCL on CIFAR10 and the other way round on CIFAR100"**
> > > >
> > > > **This only holds when both methods use weak data augmentation.** However, when **both using RandAugment**,
> > > >
> > > > |Methods|CIFAR10(60%)|CIFAR10(80%)|CIFAR100(60%)|CIFAR100(80%)|
> > > > |:---|:---|:--|:--|:---|
> > > > |RoCL: no MixUp| **92.98** | **88.18** | **69.72** | **58.72** |
> > > > |RoCL: original version| 92.82 | 88.00 | 66.79 | 54.22 |
> > > > |MentorMix: +RandAugment| 85.45 | 20.68 | 52.70 | 8.02 |
> > > >
> > > > One can see large gaps between the two methods in this case: RoCL significantly outperforms MentorMix on all the four settings.
> > > >
> > > > Note that on CIFAR10, **even with 60% wrong labels, RoCL can achieve about 93% test accuracy, which is very close to  94-95% achieved on all-correct labels!**
> > > >
> > > > On CIFAR100 with **80% noisy (wrong) labels, RoCL improves the previously known best test accuracy by 17.5%, much larger than the 2-3% improvement** on CIFAR100 reported by Cubuk et al in their RandAugment paper.

---

### Public Comment · ~Ehsan_Amid1 · 2020-11-10
**Please consider referencing/comparing to these more recent works**

I would like to point out that our work (Amid et al. 2019a) extends the Generalized CE loss (Zhang and Sabuncu 2018) by introducing two temperatures t1 and t2 which recovers GCE when t1 = q and t2 = 1. Our more recent work, called the bi-tempered loss (Amid et al. 2019b) extends these methods by introducing a proper (unbiased) generalization of the CE loss and is shown to be extremely effective in reducing the effect of noisy examples. Please consider referencing/comparing to these papers.

(Amid et al. 2019a) Amid et al. "Two-temperature logistic regression based on the Tsallis divergence." In The 22nd International Conference on Artificial Intelligence and Statistics (AISTATS), 2019.

(Amid et al. 2019b) Amid et al. "Robust bi-tempered logistic loss based on Bregman divergences." In Advances in Neural Information Processing Systems (NeurIPS), 2019.

---

> ### Author Response · Authors · 2020-11-14
> **Thanks! We will cite and discuss your papers. Here is a summary of the difference.**
>
> Hi Ehsan,
>
> Thanks for your comments! We will cite and discuss your work in the revision. Here is a summary of the differences.
>
> Although there are two temperature parameters in your two papers and our RoCL method, they control different quantities in the learning algorithms and the algorithms manipulate them in different ways.
>
> (1) In your work, the two temperatures are applied to the log loss and exponents in the softmax, i.e., they are applied respectively to the ground truth and predicted probabilities. By contrast, the two temperatures in RoCL are applied to the two data-sampling probabilities produced by the two data-selection criteria.
>
> (2) In RoCL, we gradually change the values of the two temperatures to form a preferred curriculum and their values are coupled to enable the trade-offs discussed in Section 3.1.

---

### Author Response · Authors · 2020-11-14
**To all Reviewers: A complete ablation study and full review response forthcoming ...**

Thanks very much for your review! We are working on the ablation study experiments to include in the next version. We will have a full response to your review soon, as well as the next version of the paper addressing questions from all reviewers. Please stay tuned! :-)

---

### Author Response · Authors · 2020-11-23
**Summary of Changes in Revision**

We would like to thank all reviewers for their efforts in the reviewing process. We uploaded a revision of the paper and here is a summary of the major changes:

(1) We conducted a thorough **ablation study covering 10 variants of RoCL** applied to CIFAR10/100 under two noise rates, i.e., 60% and 80%. These variants cover all the ablation studies requested by all reviewers. We reported their final accuracy in Table 5 and how their test accuracies change during training in Figure 7-10 of the Appendix. We gave a brief summary of our conclusions about the ablation study in the main paper (at the end of Section 4) and a detailed analysis in Appendix (at the end of Appendix).

(2) We added **a detailed discussion of hyperparameter settings** in Section 4 and explained how the hyperparameters were chosen according to the principle of curriculum design in RoCL.

(3) We modified Eq. (9) and made it easier to understand. We also provided an illustration to Eq. (9) in Figure 6.

(4) We removed the poor "abuse of notations" in Eq. (7)-(8).

(5) We added citations to the papers suggested by the reviewers.

(6) We made Figure 1 larger and modified other parts of paper according to all reviewers' comments.

(7) On **all** the commonly used benchmarks in our paper, **RoCL achieves new SoTA performance and outperforms the previous SoTA by a large margin, e.g., +17.5% test accuracy on CIFAR100 with 80% uniform noise.**

Please let us know any further questions. We are happy to respond.

---

### Decision · Program_Chairs · 2021-01-07
**Final Decision**

**Decision:**

Accept (Poster)

**Comment:**

This paper has been thoroughly evaluated by four expert reviewers and it had received one public comment. The authors provided extensive explanations and added technical updates to the contents of their submission in response to constructive critiques from the reviewers. Even though some minor issues have not been fully resolved in the discussion between the authors and the reviewers, I consider this paper worthy of inclusion in the program of ICLR 2021 since, albeit marginally, the apparent strengths outweigh its outstanding limitations.